# Spin-Peierls instability of the U(1) Dirac spin liquid

Urban F. P. Seifert[1,5,6] ✉, Josef Willsher [2,3,6] ✉, Markus Drescher [2,3], Frank Pollmann[2,3] & Johannes Knolle [2,3,4]

Quantum fluctuations can inhibit long-range ordering in frustrated magnets and potentially lead to quantum spin liquid (QSL) phases. A prime example are gapless QSLs with emergent U(1) gauge fields, which have been understood to be described in terms of quantum electrodynamics in 2+1 dimension (QED$_3$). Despite several promising candidate materials, however, a complicating factor for their realisation is the presence of other degrees of freedom. In particular lattice distortions can act to relieve magnetic frustration, precipitating conventionally ordered states. In this work, we use field-theoretic arguments as well as extensive numerical simulations to show that the U(1) Dirac QSL on the triangular and kagome lattices exhibits a weak-coupling instability due to the coupling of monopoles of the emergent gauge field to lattice distortions, leading to valence-bond solid ordering. This generalises the spin-Peierls instability of one-dimensional quantum critical spin chains to two-dimensional algebraic QSLs. We study static distortions as well as quantum-mechanical phonons. Even in regimes where the QSL is stable, the singular spin-lattice coupling leads to marked temperature-dependent corrections to the phonon spectrum, which provide salient experimental signatures of spin fractionalisation. We discuss the coupling of QSLs to the lattice as a general tool for their discovery and characterisation.

The presence of many competing classical ground states in frustrated magnets in conjunction with strong quantum fluctuations may stabilise quantum spin liquids (QSL) as exotic states of quantum matter. They fall outside Landau's paradigm of symmetry-breaking order, and are instead characterised by a rich entanglement structure described in terms of emergent deconfined gauge theories[1–3]. How such quantum phases can be realised in two-dimensional frustrated magnets remains an open question—definitive experimental evidence of a QSL remains outstanding despite remarkable progress in the experimental identification of many candidate materials in recent years. Most pertinently, many candidates undergo magnetic and structural ordering transitions at low temperatures which relieve magnetic frustration.

A paradigmatic example of the marked impact of spin-lattice coupling on strongly fluctuating quantum magnets is the spin-Peierls instability in 1D[4,5]. The spin-1/2 antiferromagnetic (AFM) Heisenberg chain has a spin-liquid ground state with algebraically decaying spin correlations. The coupling of the staggered spin dimerisation $(-1)^i \vec{S}_i \cdot \vec{S}_{i+1}$ to a finite, alternating lattice dimerisation produces a non-analytic energy gain by opening a spin gap. This competes with the harmonic elastic energy cost of the lattice distortion, giving the following effective potential for the distortion field $u$:

$$E(u) - E_{\mathrm{GS}} = Ku^2 - c(gu)^\chi. \tag{1}$$

[1]Kavli Institute for Theoretical Physics, University of California, Santa Barbara, CA, USA. [2]Technical University of Munich, TUM School of Natural Sciences, Physics Department, Garching, Germany. [3]Munich Center for Quantum Science and Technology (MCQST), Schellingstr. 4, München, Germany. [4]Blackett Laboratory, Imperial College London, London, United Kingdom. [5]Present address: Institute for Theoretical Physics, University of Cologne, Cologne, Germany. [6]These authors contributed equally: Urban F. P. Seifert, Josef Willsher. ✉e-mail: urban.seifert@uni-koeln.de; joe.willsher@tum.de

Given that for the isotropic Heisenberg AFM $\chi = 4/3$ and $c > 0$, the spin liquid is unstable for any finite coupling $g > 0$ and the ground state is a dimerised valence-bond solid (VBS) state with $u > 0$[6,7], as shown in the top panel of Fig. 1. Such spin-Peierls instabilities have been experimentally observed in a number of (quasi-)one-dimensional spin-chain compounds, such as $CuGeO_3$[8].

In this work we investigate spin-Peierls-type instabilities in higher-dimensional frustrated quantum magnets, with previous works having focused on degenerate ground states in 2D[9,10] or intrinsic nesting instabilities of spinon Fermi surfaces in 3D[11,12]. Motivated by the paradigmatic algebraic spin chain instability in 1D, we study the crucial question concerning the stability of (otherwise intrinsically stable) gapless QSL ground states to lattice distortions. Importantly, the absence of a gap will by no means automatically lead to an instability. For example in the $Z_2$ QSL with gapless spinons, the gauge field is always gapped, and thus at low energies the spinons are effectively non-interacting Dirac fermions. Because these have a vanishing fermion density of states, $Z_2$ QSLs can be expected to be stable up to some finite critical lattice coupling $g_c > 0$[10,13]. An open question is whether an instability might occur in U(1) Dirac spin liquids, which have been proposed as the stable, gapless ground state of frustrated Heisenberg AFMs on both the kagome[14–16] and triangular lattices[17–19]. These are described by a strongly interacting field theory of fermions in the presence of an emergent compact U(1) gauge field, which naturally represents a 2D analogue of the 1D Heisenberg AFM[20].

Here, we surprisingly uncover that under lattice coupling, U(1) Dirac QSLs (DSL) behave much more like their gapless spin-chain counterparts than previously studied 2D QSLs. That is (unlike 2D free fermion systems or $Z_2$ QSLs) infinitesimal spin-lattice couplings destroy the U(1) DSL on the triangular and kagome lattices through a spin-Peierls instability, as shown in Fig. 1. In fact, some triangular lattice candidate materials exhibit magnetic ordering transitions at low temperature that seem to coincide with structural distortions[21,22]. In order to study the effect of 2D spin-lattice couplings on the U(1) DSL in detail, we make use of its low-energy effective continuum description. This is given by quantum electrodynamics in 2+1 dimensions (QED₃), which is believed to flow to a conformally-invariant fixed point[23]. The most relevant operators at this fixed point are monopole operators, or instantons, that tunnel quanta of magnetic flux of the emergent U(1) gauge field, which can be identified with fluctuating 120-degree AFM and VBS order parameters[20,24,25]. Much like in spin chains, we show that the algebraic correlations of the monopoles are responsible for a lattice instability; understanding their symmetry properties and scaling dimension allows us to predict the ordering pattern of the lattice and spins, and to calculate the energetic exponent $\chi$ [as for 1D in Eq. (1)]. We go on to confirm the presence of an instability numerically on the triangular lattice, and provide new theoretical predictions for the regime of stability of the U(1) DSL against dynamical phonons[26].

We suggest that this improved understanding of spin-lattice coupling for algebraic QSLs may facilitate their experimental identification via thermodynamic properties of the ordering transition, or through spectroscopic signatures of the proximate QSL phase. We show that the power-law Kohn anomaly of the phonon spectrum provides a direct experimental probe of proximate DSL physics. Moreover, taking into account the phonon spectrum will be crucial for identifying novel candidate materials. Beyond the example of the 2D DSL, spin-lattice coupling may be an important mechanism for generating spin-Peierls phases of other quantum many-body systems described by emergent strongly coupled gauge theories.

## Results
### Spin-lattice coupling

We first consider static displacement fields by neglecting any intrinsic quantum dynamics, often referred to as the adiabatic limit[27,28]. We focus on the DSL state on the triangular and later generalise to the kagome lattice. Here, in the spirit of generality, we first use a symmetry-based field-theory approach which is agnostic to specific microscopic models as long as spin rotation and lattice symmetries (translations $T_1$, $T_2$, $C_6$ rotations, parity $\mathcal{P}$ and time reversal $\mathcal{T}$) are preserved. This leverages the continuum field-theoretic formulation of QED₃ in Euclidian spacetime with coordinates $x = (\tau, \mathbf{r})$. We later study a microscopic model for spin-lattice couplings and test our findings using numerical methods.

An in-plane distortion of the lattice $\mathbf{R}_i \rightarrow \mathbf{r} = \mathbf{R}_i + \mathbf{u}(\mathbf{r})$ can be described by the classical field $\mathbf{u}(\mathbf{r})$. Note that here, we are using Eulerian coordinates[29] in which the displacement field is implicitly defined with respect to global (lab) coordinates (rather than with respect to the undistorted lattice) to preserve locality, ensuring a well-defined continuum limit. The Fourier components of the distortion field are given by $\mathbf{u_Q} = V^{-1}\int d^2\mathbf{r}\, \mathbf{u}(\mathbf{r})e^{-i\mathbf{Q}\cdot\mathbf{r}}$. By symmetry, we find that there is an allowed coupling between fluctuating VBS order parameters, represented by monopoles $\Phi_a(x)$ in the QED₃ field theory (with $a = 1, 2, 3$), and the longitudinal projections $u_a^* = is_a\mathbf{X}_a \cdot \mathbf{u}_{\mathbf{X}_a}^*$ of the distortion field with lattice momenta $\mathbf{X}_a = -\mathbf{K}_a/2$ and appropriate signs

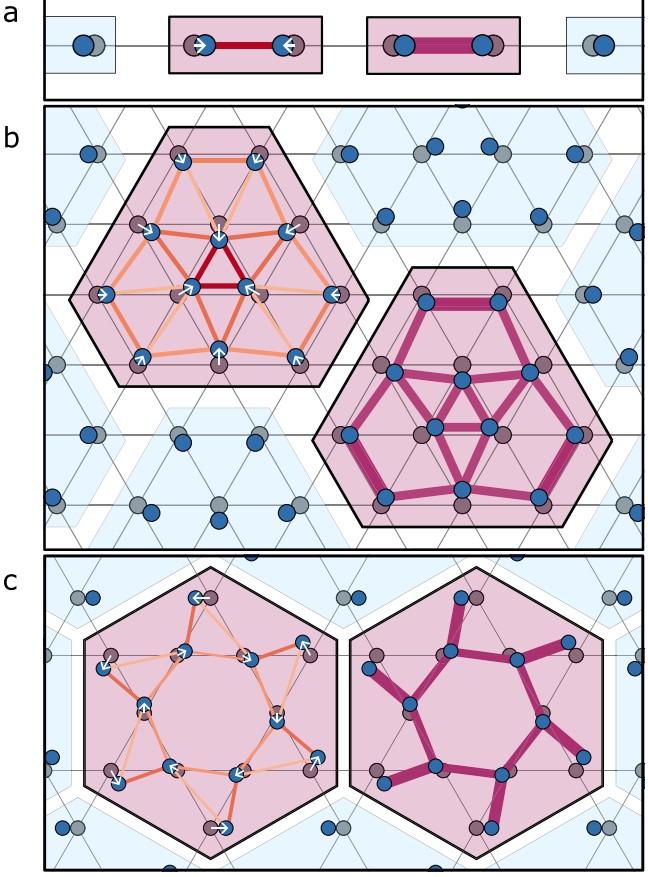

**Fig. 1 | Spin-Peierls distortions of unstable gapless spin liquids in one and two dimensions. a** In one dimension, the spin-half antiferromagnetic Heisenberg chain is unstable to dimerisation when coupled to the lattice. The U(1) Dirac spin liquid is unstable to $\sqrt{12} \times \sqrt{12}$ and pinwheel valence-bond order on the **b** triangular and **c** kagome lattices, respectively. The blue shifted dots show the distorted lattice, the vector displacements inside the unit cell $\mathbf{u}(\mathbf{r})$ are highlighted by white arrows. The enhanced nearest-neighbour bond strengths within each plaquette are depicted in the left panel, and the short-range spin-spin correlations $\langle \vec{S} \cdot \vec{S} \rangle$ are shown on the right. Structural transitions can be measured thermodynamically, and the corresponding lattice and valence-bond order can be measured with inelastic X-ray and neutron spectroscopy.

$s_a = +1, +1, -1$. The corresponding action reads (see "Methods")

$$\mathcal{S}_g[\vec{u}] = g \sum_a \int d^3x [u_a^* \Phi_a(x) + \text{h.c.}], \qquad (2)$$

where $\vec{u} = (u_1, u_2, u_3)^\top$. In general, the energy cost associated with lattice distortions can be approximated (to lowest order) by a harmonic term $\mathcal{H}_{\text{ph}}[\vec{u}] = \kappa \sum_a |u_a|^2$ where $\kappa$ is proportional to the lattice stiffness $K$.

We emphasise that, while the monopoles are strongly relevant operators in the QED$_3$ CFT and the coupling to the distortion field is symmetry-allowed, this does not necessarily imply an instability of the DSL, given the intrinsic elastic energy cost associated with a lattice distortion. For an instability to occur, the system must dynamically generate such a distortion, with the energy gain due to VBS ordering outcompeting the elastic energy cost. We will now analyse this competition using field theory and numerical methods.

### Weak-coupling spin-Peierls instability
We calculate the perturbative energy gain of the U(1) DSL coupled to lattice deformations at wavevectors $\mathbf{X}_a$, using conformal perturbation theory in the limit of small coupling $g$ with a finite-temperature regulator $\beta = T^{-1}$ (see "Methods"). The effective free energy as a function of distortion $\vec{u}$, including the energy cost due to lattice stiffness, can be written as

$$\begin{aligned} \mathcal{H}_{\text{eff}}[\vec{u}] &= \mathcal{H}_{\text{ph}}[\vec{u}] + \left( \mathcal{E}_{\text{QED}_3}[\vec{u}] - \mathcal{E}_{\text{QED}_3}[0] \right) \\ &= \left( \kappa - c_{\Delta_\Phi} g^2 \beta^{3-2\Delta_\Phi} \right) |\vec{u}|^2 + \mathcal{C}_1(\beta)|\vec{u}|^4 + \mathcal{C}_2(\beta)|\vec{u} \cdot \vec{u}|^2 + \dots \end{aligned} \qquad (3)$$

where we have introduced the numerical constant $c_{\Delta_\Phi} = 2\pi/[(\Delta_\Phi - 1)(3 - 2\Delta_\Phi)] > 0$ which is positive when the scaling dimension of the monopoles is in the range $1 < \Delta_\Phi < 3/2$, as satisfied by the predicted value $\Delta_\Phi \approx 1.02$[30]. This perturbative result is controlled for $g^2 \beta^{3-2\Delta_\Phi} |\vec{u}|^2 \ll 1$, and this functional may be understood analogous to a Ginzburg–Landau expansion around small $|\vec{u}|$. We take the constants $\mathcal{C}_1(\beta)$ and $\mathcal{C}_2(\beta)$ to be positive (cf. Supplementary Note 1).

In the zero-temperature limit $T = \beta^{-1} \to 0$, the quadratic effective energy functional (3) inverts at small coupling $g > 0$. It is therefore energetically preferable for the system to acquire a lattice distortion and form VBS order. The critical temperature scale $T_{\text{SP}} = 1/\beta_c$ for the ordering instability is obtained by analysing where the quadratic potential in $\mathcal{H}_{\text{eff}}[\vec{u}]$ [Eq. (3)] changes sign at small $|\vec{u}|$, yielding

$$T_{\text{SP}} \sim (g^2/\kappa)^{1/(3-2\Delta_\Phi)}. \qquad (4)$$

We conclude the main result of our work: the coupling of spin-singlet monopoles (acting as fluctuating VBS order parameters) to longitudinal displacement modes with wavevectors $\mathbf{X}_a$ induces a weak-coupling instability of the DSL ground state at sufficiently low temperatures for any finite coupling $g > 0$.

What is the nature of the low-temperature phase $T < T_{\text{SP}}$? Given $\mathcal{C}_{1,2}(\beta) > 0$, we infer that the ordered state will have a finite distortion satisfying $|\vec{u}| = u_0$ and $\vec{u} \cdot \vec{u} = 0$. Working at the level of (3), there is a continuous SO(3) degeneracy of possible ground states. Taken at face value, the Mermin–Wagner theorem would prohibit a thermodynamic ordering transition (unlike the one-dimensional spin-Peierls transition, where only a discrete $Z_2$ symmetry is broken). However, this continuous symmetry is only an emergent SO(3) valley symmetry at the CFT fixed point and is not present in underlying microscopic systems. Hence, anharmonic terms $\tilde{\lambda}(u_1 u_2 u_3 + \text{c.c.})$ will break the continuous symmetry back down to the discrete $C_3$ symmetry of the lattice. These terms are either generated by a dangerously irrelevant operator[31,32] in the field theory, or arise from intrinsic interactions between phonons beyond the harmonic approximation. The resulting energy-minimising

configurations for $\tilde{\lambda} < 0$ are given by

$$\vec{u}_0 = [u_0/\sqrt{3}]\left( 1, e^{2\pi i/3}, e^{-2\pi i/3} \right)^\top. \qquad (5)$$

In real-space, the pattern is shown in Fig. 2b. The spins react to the lattice breaking the translation and $C_6$ rotation symmetries by forming $\sqrt{12} \times \sqrt{12}$ VBS patterns with $C_3$ symmetry (cf. Supplementary Note 1)[33]. In the field-theory picture, the choice of $\vec{u} = \vec{u}_0$ determines the ordering of the monopole excitations through minimising the coupling Eq. (2) as $\text{sign}(g)\langle\vec{\Phi}\rangle \sim -\vec{u}_0$. Using the leading-order mapping between VBS monopoles and nearest-neighbour spin-dimer correlations [given by (17) in "Methods"], we can evaluate the predicted nearest-neighbour spin-spin correlation function

$$\langle \vec{S}_{\mathbf{r}_i} \cdot \vec{S}_{\mathbf{r}_i + \boldsymbol{\delta}_a} \rangle \simeq s_a \text{Re}(\langle\Phi_a\rangle e^{i\mathbf{X}_a \cdot \mathbf{r}_i}). \qquad (6)$$

The resulting lattice distortion and (leading-order) VBS order from this analysis are shown in Figs. 2b and e. We emphasise that the computed patterns above predict the symmetry of the distorted lattice, but the precise strength of $\langle \vec{S}_i \cdot \vec{S}_j \rangle$ on a given bond is not accessible within our field-theoretic approach and is expected to depend on microscopic details. We present analogous results for the kagome lattice in Supplementary Note 2.

### Instability of the AFM Heisenberg model
We now go beyond field-theoretic arguments and consider the $J_1$–$J_2$ triangular lattice AFM Heisenberg model, for which numerical simulations suggest a U(1) DSL ground state[34,35]. This microscopic model allows us to explicitly study the system's energetic response to lattice distortions and compare with $\Delta\mathcal{E}_{\text{QED}_3}[\vec{u}]$ as constructed above. To this end, we assume that the couplings $J_{ij}$ between two sites are homogenous and decreasing functions with distance $\mathbf{r}_i - \mathbf{r}_j$, and we consider a simple exponential form $J_{ij} \sim J e^{-|\mathbf{r}_i - \mathbf{r}_j|/\alpha}$. In general, we expect that the first derivative of $J_{ij}$ as a function of distance is non-vanishing, which implies that for small distortions one may linearise. Then, the change in bond length modifies the interaction $J_1$ between nearest-neighbour unit vectors $\hat{\boldsymbol{\delta}}_a$ as

$$H_\alpha[\mathbf{u}(\mathbf{r}_i)] = \sum_i \sum_{a=1,2,3} J_1 \left[ 1 + \alpha \hat{\boldsymbol{\delta}}_a \cdot (\mathbf{u}(\mathbf{r}_i) - \mathbf{u}(\mathbf{r}_i + \boldsymbol{\delta}_a)) \right] \vec{S}_{\mathbf{r}_i} \cdot \vec{S}_{\mathbf{r}_i + \boldsymbol{\delta}_a}, \qquad (7)$$

as constructed in "Methods". Here, the (dimensionful) coefficient $\alpha$ is some constant of proportionality that arises upon linearising and characterises the degree of spin-lattice coupling in the microscopic Heisenberg model. We consider four distinct distortion patterns and perform DMRG simulations on circumference-$L$ cylinders (see "Methods"). Instead of using $\beta$ as a regulator, our simulations effectively take the zero-temperature limit while working on a finite (cylindrical) geometry with circumference $L$, which acts as an IR cutoff instead. In this case, the weak-coupling energy response to distortions $u_a$ at the wavevectors $\mathbf{X}_a$ is given by $\Delta\mathcal{E}_{\text{QED}_3}[\vec{u}] = -\tilde{c}_{\Delta_\Phi} g^2 L^{3-2\Delta_\Phi} |\vec{u}|^2$, with some constant $\tilde{c}_{\Delta_\Phi}$. The instability upon coupling to lattice distortions at wavevectors $\mathbf{X}_a$ is reflected in the IR limit by the growth in amplitude of the energy response; $\mathcal{E}_{\text{QED}_3}[\vec{u}]$ grows as a power of the system size, which is controlled by the monopole scaling dimension[36]. In contrast, we expect that distortions at other momenta, e.g., the $\mathbf{K}$ or $\mathbf{M}$ points, will generically produce a finite response that is independent of system size as $L \to \infty$.

In Fig. 2a, we compare the energy gain of the spin system under static lattice distortions on the $L = 6$ cylinder. We compare momenta at symmetric points in the Brillouin zone to the 12-site distortion pattern in Fig. 2b. We model the energy gain as $-\Delta\mathcal{E}[\delta] = A_{\mathbf{Q}}(L)\delta^2$, where $\delta = \alpha|\vec{u}|$ is a dimensionless distortion parameter that measures

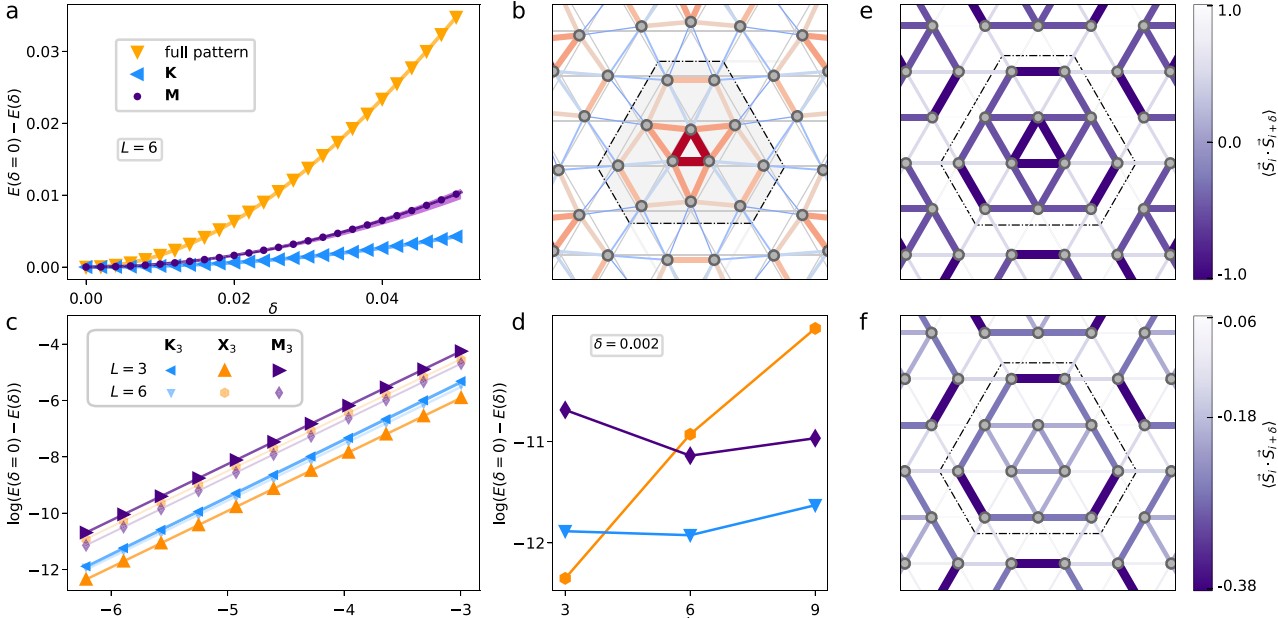

**Fig. 2 | Numerical observation of the spin-Peierls instability of the triangular-lattice AFM Heisenberg model.** Density matrix renormalisation group calculations of the next-nearest-neighbour frustrated triangular lattice model ($J_2/J_1 = 1/8$) are presented, where the undistorted lattice displays a gapless spin-liquid ground state. **a** Energy gain of the spin system under various simulated lattice distortions on the $L = 6$ circumference cylinder. Patterns considered are generated by momenta $\mathbf{K}_3$, and $\mathbf{M}_{1,2,3}$ (here the purple shaded region represents the range of responses for the three $\mathbf{M}_a$, showing minimal dependence on cylinder orientation; see Supplementary Note 3), as well as the full 12-site distortion, defined by Eq. (5). **b** Nearest-neighbour exchanges under the 12-site unit cell distortion; red/blue bonds show enhanced/suppressed bonds. **c** System-size dependence of energy gain for $L = 3, 6$. We compare three distinct distortion patterns generated by the momenta $\mathbf{K}_3$, $\mathbf{M}_3$, and $\mathbf{X}_3$ which are compatible with the cylinder geometry. Logarithmic axes show the quadratic energy gain for each pattern. **d** $L$-dependence of the energy gain for the data point $\delta = 0.002$ for cylinders with circumference of up to $L = 9$. The colours indicate the same distortion patterns as in **c**. Only the distortion at $\mathbf{X}_3$ shows strong $L$-dependence, which confirms that a weak-coupling instability of the U(1) Dirac spin liquid is realised in this numerical study of the $J_1$–$J_2$ model. **e,f** The spin-spin correlation function on dimers $\langle \vec{S}_i \cdot \vec{S}_j \rangle$ in the presence of a 12-site lattice distortion **b**. Subfigure **e** shows the theoretical prediction from the leading order analysis of the symmetry of operators in the CFT spectrum (on an arbitrary scale); and **f** is the numerical calculation performed with DMRG, evaluated with distortion parameter $\delta = 0.05$ and $L = 6$ and averaged over orientations relative to the cylinder axis.

the change in bond strength [n.b. we define it as such since the Hamiltonian Eq. (7) only depends on the product]. All curves are fully consistent with the predicted quadratic behaviour at small distortions, and we can see that the 12-site unit cell pattern shows the strongest response amplitude, which we explain as being due to the coupling to commensurate VBS fluctuations (equivalently, the spin-singlet monopoles). Figure 2f shows the nearest-neighbour spin-spin correlations $\langle \vec{S} \cdot \vec{S} \rangle$ of the symmetric 12-site pattern DMRG, compared to the translationally-invariant correlations (with value $-0.18$) of the spin-disordered DSL ground state in the undistorted model. The variation of singlet correlations in the distorted model shows good agreement relative to the prediction based on the monopole transformation properties in Fig. 2e where notably we see that the correlations are enhanced on all shortened nearest-neighbour bonds. The difference in the exact distribution of weight inside the unit cell is either attributable to neglecting contributions to the nearest-neighbour correlator by multi-spin operators, or errors introduced due to the use of an anisotropic $L = 6$ cylinder geometry. In conclusion, we find that the explicitly enforced 12-site lattice distortion leads to a strong energetic gain and transition to a short-range correlated phase, which is consistent with a DSL−VBS transition on a finite-circumference cylinder.

Now we turn to the $L$-dependence of the energy response in the weak-coupling regime as a signature of an instability. To do this we calculate the energy gain on $L = 3, 6, 9$ cylinders for three patterns that are compatible with the boundary conditions. We find no significant increase of the energy response for $\mathbf{M}_3$ and $\mathbf{K}_3$ distortions; in contrast, the distortion $\mathbf{X}_3$ shows significant dependence on system-size, as plotted in Figs. 2c, d (cf. Supplementary Note 3 for computational

details), strongly suggestive of the IR-divergence predicted from our field-theoretic analysis. We conclude that the numerical simulations performed on finite-circumference cylinders are compatible with our predictions that the instability (in large systems) will be dominated by distortions at the spin-singlet monopole wavevectors $\mathbf{X}_a$. We suggest this is a signature of the weak-coupling lattice instability of the U(1) DSL ground state.

## Strong-coupling behaviour

In the previous sections, we have seen that perturbing the CFT fixed point with a monopole-lattice coupling leads to infrared divergences which must be regulated assuming a finite system size $L < \infty$ (or equivalently finite temperature $T > 0$). When the breakdown of perturbation theory $g^2 L^{3-2\Delta_\Phi} |\vec{u}|^2 \gg 1$ signals an instability, the system's response to deforming the CFT fixed point in the thermodynamic limit (i.e., at $T = 0$ and $L \to \infty$) is expected to lead to a critical (non-trivial) power-law response, with critical exponents determined by scaling dimensions of operators at the fixed point (this is also understood from the fact that at $T = 0$, $L \to \infty$ the monopole-lattice coupling constitutes the only energy scale in the otherwise dimensionless theory−conversely, in the weak-coupling limit, the IR regulator introduces a second scale). Therefore, we use below a scaling Ansatz which infers the CFT's response to the distortion from scaling and symmetry arguments. This approach will allow us to generalise the 1D energy response from Eq. (1) to the 2D model, as well as predicting ground-state properties like the scaling form of the induced gap size.

From power-counting of Eq. (2) with the QED₃ fixed point theory, one may write down a strong-coupling energy density that is scale-

invariant and compatible with the symmetries of the field theory

$$\mathcal{H}_{sc}[\vec{u}] = \kappa|\vec{u}|^2 - c_{sc}|g\,\vec{u}|^\chi \tag{8}$$

with $\chi = 3/(3-\Delta_\Phi) < 2$ and $c_{sc} > 0$ an undetermined positive scaling constant[29]. This non-analytic contribution to the ground-state energy must outcompete the (leading) quadratic energy cost for the longitudinal displacements. We find it to be energetically preferable for the system to undergo a lattice distortion with amplitude $u_0 \neq 0$ as long as $\Delta_\Phi < 3/2$. We stress that here, the form (8) possesses an accidental SO(3)$_{valley}$ symmetry which is emergent at the CFT fixed point, which will be reduced to a discrete subgroup (compatible with the discrete UV symmetries of the model) by dangerously irrelevant operators or anharmonic terms, as discussed above.

In the context of our numerical study, we predict that as the thermodynamic limit is taken, the quadratic energy gain $|\delta|^2$ which grows with larger system sizes will receive corrections and, for large enough circumferences, instead be described by a different power law $|\delta|^\chi$ that is $L$-independent. Probing this non-analytic behaviour directly and extracting the coefficient $c_{sc}$ numerically would allow a concrete prediction of ground state properties of potentially unstable U(1) DSL candidate materials.

Within this scaling Ansatz, we can further estimate the magnitude of the induced displacement field by minimising $\mathcal{H}_{sc}[\vec{u}]$ and obtain

$$u_0 = \left[\frac{2}{\chi c_{sc}}\left(\frac{\kappa}{g^\chi}\right)\right]^{1/(\chi-2)}. \tag{9}$$

The corresponding gap scales as $\Delta_{gap} \sim (g^2/\kappa)^{1/(3-2\Delta_\Phi)}$, in accordance with the spin-Peierls temperature Eq. (4). We predict that the ratio $\Delta_{gap}/T_{SP}$ is system-independent and would be universal for all U(1) DSL materials which exhibit a spin-Peierls instability in the ground state.

Given that the above discussion is rather general in the sense that it only relies on the existence of a symmetry-allowed coupling between the distortion field and a relevant operator in the CFT, a corollary of our strong-coupling approach follows: one may tentatively extend our results for the DSL to more general couplings between gapless deconfined phases of matter (described by a CFT) to lattice distortions in any dimension, as in Supplementary Note 4. The criterion for the spin-Peierls instability to occur is that there is a VBS-order parameter, expressed as an instanton of an underlying gauge theory, with scaling dimension $\Delta < d/2$ (note that this is stronger than relevance $\Delta < d$). Indeed, this criterion corresponds to the collapse of quantum criticality upon identifying the critical exponents $\nu^{-1} = d - \Delta$ and $z = 1$[37], and reproduces the regime where antiferromagnetic XXZ chains suffer a spin-Peierls instability (namely, spin-anisotropy $0 \geq \Delta_{aniso} \geq 1$[5]). It will be interesting to study the physical consequences in detail for deconfined quantum critical points[32,38,39] or non-Lagrangian QSL phases[40].

## Dynamical phonons

Next, we go beyond the assumption of adiabatic distortions by promoting the corresponding relevant components of the distortion field to be dynamical degrees of freedom $u_a \rightarrow u_a(\tau, \mathbf{r})$ (we work in imaginary time/Euclidean signature). Such phonon dynamics introduces an additional finite energy scale such that the U(1) DSL may remain stable even at zero temperature, and conversely the phonon-monopole coupling will lead to marked signatures in the phonon spectral function. While we explicitly treat in this section a model of non-dispersive (optical) phonons, we note that this is also an appropriate model for generic phonon bands away from the $\mathbf{\Gamma}$-point, as we are explicitly interested at small momenta (compared to the inverse lattice spacing) around the finite lattice momentum of the singlet monopoles $\mathbf{X}_a$.

We add dynamics for the long-wavelength fluctuations of the $\mathbf{X}_a$-displacement modes via a kinetic energy term, $\mathcal{S}_{kin} = \rho\sum_a|\partial_\tau u_a(\tau, \mathbf{r})|^2$,

where $\rho$ corresponds to a microscopic phonon mass density. The physical phonon modes have energy $\omega_0 \equiv \sqrt{\kappa/\rho}$ at the momentum $\mathbf{X}_a$; we approximate this as an optical phonon with constant energy[5,41], governed by the action

$$S_{ph}[\vec{u}(\tau,\mathbf{r})] = \int d\tau \int d^2\mathbf{r}\left[\rho|\partial_\tau\vec{u}(\tau,\mathbf{r})|^2 + \kappa|\vec{u}(\tau,\mathbf{r})|^2\right]. \tag{10}$$

The full action of the system then reads $\mathcal{S}[u_a(x)] = \mathcal{S}_{ph}[u_a(x)] + \mathcal{S}_{QED_3} + S_g$ where $\mathcal{S}_{QED_3} = \int d^3x \mathcal{L}_{QED_3}$ is the fixed-point action for QED$_3$, and $S_g$ is the coupling between monopoles and (now dynamical) phonons Eq. (2). The phonon Green's function from Eq. (10) is

$$\langle u_a(x)u_b^*(y)\rangle_{ph} = \delta_{a,b}G(x-y) = \delta_{a,b}\frac{\delta^{(2)}(\mathbf{r}_x - \mathbf{r}_y)}{2\rho\omega_0}e^{-\omega_0|\tau_x-\tau_y|}. \tag{11}$$

As long as the phonons remain gapped $\omega_0 > 0$, these degrees of freedom can be integrated out exactly to obtain a retarded interaction between QED$_3$ degrees of freedom, giving $S = \mathcal{S}_{QED_3} + \mathcal{S}_{\Phi\Phi}$, where

$$\mathcal{S}_{\Phi\Phi} = -g^2\sum_a\int d^3x\,d^3y\,\Phi_a^\dagger(x)G(x-y)\Phi_a(y). \tag{12}$$

In the adiabatic limit $\omega_0 = 0$, the monopole-monopole interaction in (12) becomes temporally non-local, implying that modes become correlated at large separations and leading to an instability at infinitesimal couplings as per our earlier treatment. In the opposite, antiadiabatic, limit $\omega_0 \rightarrow \infty$ ($\kappa$ constant), the Green's function becomes purely local, $G(x-y) = (1/2\kappa)\delta^{(3)}(x-y)$ and the DSL remains stable. Considering both limits, it becomes clear that there must exist a transition at some intermediate coupling $g_c$ determined by $\omega_0$. A scaling analysis using the exponential form of $G(\tau_x - \tau_y)$ in Eq. (11) shows that the U(1) DSL is stable for weak-interaction strengths $g < g_c$ set by a power-law of $\omega_0$, as

$$g_c^2 \sim \kappa\,\omega_0^{3-2\Delta_\Phi}. \tag{13}$$

This result is confirmed by a perturbative calculation in Supplementary Note 5.

We equivalently investigate how the phonon-monopole coupling is manifested in spectral properties of the phonon by instead integrating out the fluctuations of QED$_3$ perturbatively. Working at second order in perturbation theory, we find a correction to the propagator of the phonon mode $u_a$, given by

$$G_a^{-1}(\omega) = \rho\omega^2 + \kappa - g^2\langle\Phi_a^\dagger\Phi_a\rangle_{QED_3}(\omega). \tag{14}$$

Continuing to real frequencies, this implies that the phonon quasiparticle dispersion $\omega_a$ becomes renormalised and is determined by solutions to the implicit equation

$$\omega^2 = \omega_0^2 - g^2\rho^{-1}\chi_a'(\omega,T), \tag{15}$$

where $\chi'$ is the real part of the VBS susceptibility $\chi_a'(\omega,T) \equiv \text{Re}\,\chi_{VBS}(\omega,\mathbf{X}_a + \delta\mathbf{q},T)|_{\delta\mathbf{q}=0}$, which for momenta close to $\mathbf{X}_a$ is given by singular VBS monopole-monopole correlations[24] at $T = 0$.

At zero temperature $T = 0$, the VBS-susceptibility has a power-law divergence $\chi_a'(\omega,T) \sim \omega^{-(3-2\Delta_\Phi)}$ which implies a breakdown of the quasiparticle picture of the phonon. Instead, around the momenta $\mathbf{X}_a$, the phonon spectral function will display a quantum-critical continuum, a signature of deconfinement in the spin sector of the system[39], as shown in Fig. 3b. For temperatures $T \gg \omega$, there is no breakdown of the quasiparticle picture but a renormalisation of the phonon dispersion; scaling arguments imply that[42] $\chi_a'(\omega,T) \sim T^{-(3-2\Delta_\Phi)}$, and from (15) it follows that the phonon frequency

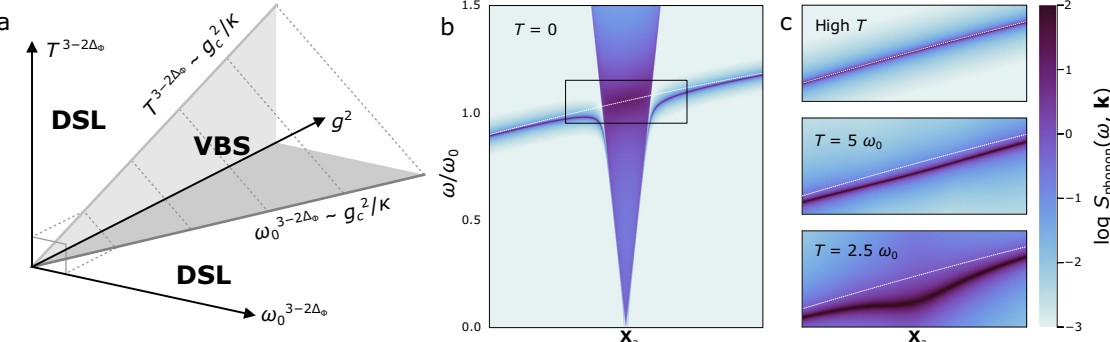

**Fig. 3 | Phase diagram and Kohn anomaly of the U(1) DSL with dynamical phonons. a** Scaling phase diagram for the spin-Peierls-VBS instability of the U(1) DSL, as a function of coupling $g$, temperature $T$ and frequency $\omega_0$. The shaded regions indicate the unstable parameter regimes, based on the finite-temperature calculation in the classical ($\omega_0 \to 0$) limit and the zero-temperature result for dynamical phonons ($\omega_0 > 0$). Phonon spectral function $\log S_{\mathrm{phonon}}(\omega, \mathbf{k})$ plotted for **b** zero and **c** finite temperature along the momentum slice between $\boldsymbol{\Gamma}$ and $\mathbf{K}_a$.

Spin-phonon coupling is set to $g = 0.3$. The phonon spectral function is approximated by extending the interacting phonon propagator $G_a(\omega) = G(\omega, \mathbf{X}_a)$ to momenta $\mathbf{X}_a + \mathbf{q}$ given a microscopic model for the bare phonon dispersion $\omega_0(\mathbf{q})$ (white dashed line). We use a heuristic scaling form for the VBS correlator at finite temperatures for illustrative purposes (see "Methods"). At finite temperature $T > T_{\mathrm{SP}}$ the phonon dispersion displays a Kohn anomaly, and at zero temperature, the quasiparticle picture breaks down.

is shifted downwards. We conclude that there exists a Kohn anomaly in the phonon dispersion[43,44], i.e., at and close to the wavevectors $\mathbf{K}_a/2$ there will be a characteristic softening of the otherwise gapped phonon depending on temperature with exponent $2\Delta_\Phi - 3$, as shown in Fig. 3c. Importantly, this provides an experimentally accessible signature of a proximate U(1) DSL through the monopole-phonon couplings.

We extend the above analysis in "Methods" to the critical regime by perturbatively calculating when the softened phonon hits zero energy and condenses, giving rise to a finite static lattice distortion. This approach recovers expressions for critical couplings $g_c^2 \sim T^{3-2\Delta_\Phi}$ in the classical (static) limit [as in Eq. (4)] and $g_c^2 \sim \omega_0^{3-2\Delta_\Phi}$ in the zero-temperature limit [as in Eq. (13)]. These two limits are seen to correspond, as the phonon frequency $\omega_0$ sets an effective temperature scale $T \sim \omega_0$ (as indicated by the dashed lines) which determines a correlation scale for critical fluctuations in the DSL. Together, we produce the phase diagram, shown in Fig. 3a.

## Discussion

Turning towards experiment, we note that several triangular-lattice candidate materials have recently been identified, such as NaYbO$_2$[45,46] and YbZn$_2$GaO$_5$[47], which exhibit broad inelastic neutron-scattering spectra and $T^2$-scaling of the magnetic specific heat at low temperatures, suggestive of Dirac-type gapless excitations. An instability of spin-Peierls type is potentially realised in the class of triangular-lattice organic Mott insulators $\kappa$-(ET)$_2$X, e.g., where $X = $Cu$_2$(CN)$_3$, which has a gapped ground state and finite structural distortion[21]. Most interestingly, the anisotropic triangular-lattice compound $\kappa$-(ET)$_2$B(CN)$_4$ has a (field-dependent) transition at 5 K[22,48] to a spin-Peierls phase with an anomalously large value of $\Delta_{\mathrm{gap}}/T_{\mathrm{SP}}$. While this phase was proposed to emerge from a spin-Peierls instability of coupled spin chains, i.e., a quasi-1D effect, our work suggests that these structural transitions in spin-lattice coupled systems could also be of an intrinsically 2D origin. It will be interesting to gain an understanding of the robustness of the spin-Peierls instability of the DSL in deformed triangular lattices (i.e., with a reduced UV symmetry group). This hypothesis could be assessed by more detailed studies that resolve the spatial structure of VBS ordering and lattice distortion in the above systems.

Our general framework and formalism can be straightforwardly applied to the U(1) DSL state on the kagome lattice[14,20,25], where the resulting ordered state is the pinwheel VBS order[49,50] of Fig. 1c (a state to which the phonon-free Heisenberg model is remarkably robust[51]). This pinwheel order has been measured in the deformed kagome-

lattice AFM Heisenberg compound Rb$_2$Cu$_3$SnF$_{12}$[50,52] using inelastic neutron scattering. We predict that U(1) spin liquids with more fermions are stable against infinitesimal lattice distortions as the monopole operator scaling dimension depends on the fermion number; this means that spin-orbital liquids are not expected to show a spin-Peierls instability[53].

Within the stable DSL, a monopole-phonon coupling is responsible for a Kohn anomaly visible in the phonon spectrum, which may provide insights into the critical correlations of a DSL, similar to elastic signatures of quantum critical points[54]. We further suggest that the singular response of the DSL to externally induced lattice displacements, in particular via strain, may provide a fruitful avenue for experimental characterisation. We hope that a better understanding of the stability of QSLs with respect to coupling to phonons or structural disorder will help for the eventual discovery of these enigmatic quantum liquids in real materials.

## Methods
### Conformal field theory

We introduce here the description of the U(1) DSL as a conformal field theory (CFT) in 2+1 dimensions[30,55–58]. The CFT is characterised by its conformal data, consisting of (1) a spectrum of scaling operators and (2) the operator product expansion (OPE). For this "Methods" section, it will suffice to focus on the lowest-lying primary operators in the CFT spectrum: for $N_f = 4$-flavour QED$_3$, these are the six charge-1 monopole operators. The gauge-invariant combination of monopole operators and two Dirac zero modes $f_\alpha^\dagger$ reads[20,25]

$$\Phi_{\alpha\beta}^\dagger \sim f_\alpha^\dagger f_\beta^\dagger \mathcal{M}_{2\pi}^\dagger, \tag{16}$$

where $\alpha$, $\beta$ are SU(4)-indices. A recent CFT bootstrap study[30] estimates the scaling dimension $\Delta_\Phi \in (1.02, 1.04)$, very similar to the large-$N$ (subleading order) result $\Delta_\Phi \approx 1.02$[58] and compatible with other works[57,59]. As written here, $\Phi_{\alpha\beta}^\dagger$ transforms in the antisymmetric rank-2 representation **6** of SU(4). It is convenient to use the isomorphism SO(6) = SU(4)/$Z_2$, such that we can take $\Phi_b^\dagger$ with $b = 1, \ldots, 6$ to transform as a six-dimensional vector. The lowest order correlation functions are evaluated exactly as $\langle \Phi_a \rangle = 0$ and $\langle \Phi_a^\dagger(x)\Phi_b(y) \rangle = \delta_{ab}|x - y|^{-2\Delta_\Phi}$. The next-lowest lying primaries are the fermion bilinears; their properties and the OPEs of the U(1) DSL are written in Supplementary Note 1.

Importantly, microscopic (lattice) UV symmetries such as translations, discrete rotations, and reflections are embedded in the enlarged symmetry group of the IR theory (in other words,

**Table 1 | Discrete symmetry transformations of monopole operators**[20,25]

|  | $T_j$ | $R$ | $C_6$ | $\mathcal{T}$ |
|---|---|---|---|---|
| $\mathcal{S}_1 = \Phi_1$ | $e^{-i\mathbf{X}_1 \cdot \mathbf{a}_j}\mathcal{S}_1$ | $-\mathcal{S}_3$ | $\mathcal{S}_2^\dagger$ | $\mathcal{S}_1^\dagger$ |
| $\mathcal{S}_2 = \Phi_2$ | $e^{-i\mathbf{X}_2 \cdot \mathbf{a}_j}\mathcal{S}_2$ | $\mathcal{S}_2$ | $-\mathcal{S}_3^\dagger$ | $\mathcal{S}_2^\dagger$ |
| $\mathcal{S}_3 = \Phi_3$ | $e^{-i\mathbf{X}_3 \cdot \mathbf{a}_j}\mathcal{S}_3$ | $-\mathcal{S}_1$ | $-\mathcal{S}_1^\dagger$ | $\mathcal{S}_3^\dagger$ |
| $\mathcal{V}_a = \Phi_{a+3}$ | $e^{-i\mathbf{K}_a \cdot \mathbf{a}_j}\mathcal{V}_a$ | $\mathcal{V}_a$ | $-\mathcal{V}_a^\dagger$ | $-\mathcal{V}_a^\dagger$ |

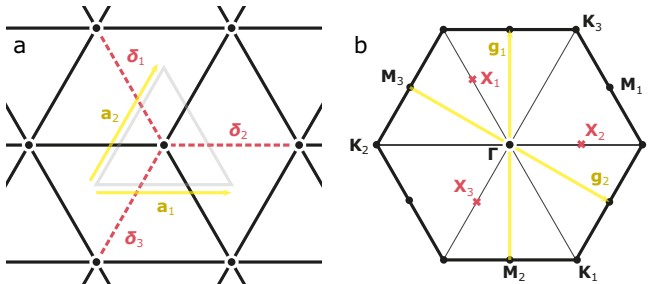

**Fig. 4 | Unit cell in real and reciprocal space. a** Triangular lattice with unit vectors $\mathbf{a}_i$ and nearest-neighbour bonds $\boldsymbol{\delta}_a$ highlighted. **b** Reciprocal lattice with inverse lattice vectors $\mathbf{g}_i$ and high-symmetry points in the Brillouin zone $\mathbf{K}_a$ and $\mathbf{M}_a$ labeled. The points $\mathbf{X}_a = -\mathbf{K}_a/2$ are the momenta eigenvalues of gapless monopole excitations[20,25].

microscopic symmetries are not broken in the flow to the IR fixed point). The monopole operators $\Phi_b$ carry non-trivial quantum numbers under lattice symmetries, which have been determined in refs. 20,25 by complementary numerical and spinon band topology-based analyses. The latter makes evident how non-trivial monopole quantum numbers follow from the distribution of gauge charges in the system's unit cell. The two (independent) microscopic symmetries $SU(2)_{\text{valley}}$ and $SU(2)_{\text{spin}}$ as subgroups of $SU(4)$ correspond to mutually commuting $SO(3)_{\text{valley}}$ and $SO(3)_{\text{spin}}$ subgroups of $SO(6)$, such that we can decompose $\Phi_b$ into valley-triplet, spin-singlet ($b = 1, 2, 3$) and valley-singlet, spin-triplet components (4, 5, 6). In Table 1, we reproduce the transformation of monopoles on the triangular lattice under discrete lattice translations $T_j$ by unit vectors $\mathbf{a}_j$, reflections $R$ in the vertical direction, discrete rotations $C_6$, and time reversal $\mathcal{T}$. The relevant (inverse) unit vectors and high-symmetry momenta are shown in Fig. 4.

The three valley-triplet, spin-singlet monopoles ($\mathcal{S}_a = \Phi_a, a = 1,2,3$) carry the same quantum numbers as order parameters of valence-bond solids on the triangular lattice with lattice momenta $\mathbf{X}_a = -\mathbf{K}_a/2$, respectively. Hence, we can write a mapping (up to a global multiplicative constant not fixed by symmetry)

$$\vec{S}_{\mathbf{r}_i} \cdot \vec{S}_{\mathbf{r}_i + \boldsymbol{\delta}_a} \simeq s_a \,\text{Re}[\mathcal{S}_a(\mathbf{r}_i)e^{i\mathbf{X}_a \cdot \mathbf{r}_i}], \tag{17}$$

where we introduce the sign factors $s_a = +1, +1, -1$. This leading-order mapping neglects possible multi-spin terms which may have the same transformation properties. The three spin-triplet monopoles ($\mathcal{V}_a = \Phi_{a+3}, a = 1,2,3$) transform identically as order parameters for antiferromagnetic 120° Néel order which determine the spin density as

$$S_i^\alpha \simeq \text{Re}[i\mathcal{V}_a(\mathbf{r}_i)e^{i\mathbf{K}_a \cdot \mathbf{r}_i}], \tag{18}$$

where $\alpha = x, y, z$ (1,2,3) denotes the three $SU(2)_{\text{spin}}$ components. These expressions gives way to the interpretation of the monopoles as disorder operators[60], with their proliferation $\langle\Phi^a\rangle \neq 0$ yielding conventionally ordered phases. The exact 2-point correlation function implies that in the U(1) DSL phase, there is diagonal quasi long-range

order of the AFM and VBS order parameters $n(x)_{\text{AFM/VBS}}$ with equal exponents $\langle n(x)n(y)\rangle_{\text{AFM/VBS}} = |x-y|^{-2\Delta_\Phi}$. Consequently, one may say that the spin-triplet/singlet susceptibility taken at the respective wavevector behaves as

$$\chi_{\text{VBS}}(\omega, \mathbf{X}_a + \mathbf{q}) = \chi_{\text{AFM}}(\omega, \mathbf{K}_a + \mathbf{q}) = [c^2|\mathbf{q}|^2 - \omega^2]^{-(3/2-\Delta_\Phi)}, \tag{19}$$

where $c$ is some emergent speed of light.

Due to their non-trivial representations under lattice transformations, individual monopoles and mass terms cannot trivially be added to the action[20]. On the triangular lattice the symmetry-allowed four-fermion interaction[61] and lowest-lying triple-monopole terms are believed to be irrelevant[58], meaning the U(1) DSL can be an intrinsically stable phase of matter.

**Lattice coupling**

Our starting point on the triangular lattice is the observation that the valley-triplet (spin-singlet) monopoles $\Phi_a$ ($a = 1, 2, 3$), which act as order parameters for VBS orders, have lattice momenta $\mathbf{X}_a = -\mathbf{K}_a/2$. Using Table 1, we note that the following deformation to the DSL fixed point is symmetry-allowed,

$$\mathcal{H} \sim (e^{i\mathbf{X}_1 \cdot \mathbf{R}}\Phi_1 + e^{i\mathbf{X}_2 \cdot \mathbf{R}}\Phi_2 - e^{i\mathbf{X}_3 \cdot \mathbf{R}}\Phi_3 + \text{h.c.}). \tag{20}$$

We emphasise that care must be taken in separating length scales: $\mathbf{R}(\mathbf{X}_a)$ are coordinates (wavevectors) on the order of microscopic length scales such as the lattice constant a, at which the $\Phi_a$ transform as given in Table 1. The scaling nature of the $\Phi_a$ as primaries in a CFT, and the corresponding power-law form of correlation functions only holds on much longer length scales (and, equivalently, sufficiently small momenta), where a low-energy continuum formulation becomes justified. While Eq. (20) is symmetry allowed, the oscillating prefactors average out on sufficiently long length scales, such that the perturbation is strongly irrelevant.

We now consider an in-plane deformation of the real-space lattice $\mathbf{R} \to \mathbf{r} = \mathbf{R} + \mathbf{u}(\mathbf{r})$, where $\mathbf{u}(\mathbf{r})$ is a displacement field, as shown in Fig. 1b. Note that we work in implicitly-defined (Eulerian) continuous coordinates $\mathbf{r}$ of the deformed system. Expanding Eq. (20) to the first non-trivial order in $|\mathbf{u}(\mathbf{r})|/a \ll 1$ (assuming that lattice distortions are small compared to the lattice constant a), we obtain the monopole-lattice coupling Hamiltonian

$$\mathcal{H}_g[\mathbf{u}(\mathbf{r})] = g \sum_{a=1,2,3} s_a \left[(i\mathbf{X}_a \cdot \mathbf{u}(\mathbf{r}))e^{i\mathbf{X}_a \cdot \mathbf{r}}\Phi_a + \text{h.c.}\right], \tag{21}$$

where we use again the sign factors $s_a = +1, +1, -1$ for convenience of notation. We have dropped the 0-th order terms (with oscillating phases) in the expansion. Note that $is_a\mathbf{X}_a \cdot \mathbf{u}$ transforms as a scalar under point-group operations $R$ and $C_6$ and is appropriately even under time reversal $\mathcal{T}$; therefore the prefactor retains the full symmetry of the undistorted system Eq. (20). We can expand the real-component distortion field $\mathbf{u}(\mathbf{r})$ in eigenstates of the lattice momentum $\mathbf{u}_\mathbf{Q}$

$$\mathbf{u}(\mathbf{r}) = \sum_\mathbf{Q} e^{i\mathbf{Q} \cdot \mathbf{r}}\mathbf{u}_\mathbf{Q}, \tag{22}$$

where reality of $\mathbf{u}(\mathbf{r})$ implies $\mathbf{u}_\mathbf{Q} = \mathbf{u}_{-\mathbf{Q}}^*$. In momentum space, the coupling to monopoles is of the form

$$\mathcal{H}_g[\mathbf{u}(\mathbf{r})] = g \sum_{a=1,2,3} \sum_\mathbf{Q} s_a \left[(i\mathbf{X}_a \cdot \mathbf{u}_\mathbf{Q})e^{i(\mathbf{X}_a + \mathbf{Q}) \cdot \mathbf{r}}\Phi_a + \text{h.c.}\right]. \tag{23}$$

Crucially, only terms with $\mathbf{Q} = -\mathbf{X}_a$ do not contain any oscillating prefactors and will thus be the ones relevant at the lowest energies; these

are

$$\mathcal{H}_g[\mathbf{u}(\mathbf{r})] = g \sum_{a=1,2,3} s_a \left[ \left( i\mathbf{X}_a \cdot \mathbf{u}^*_{\mathbf{X}_a} \right) \Phi_a + \text{h.c.} \right]. \tag{24}$$

Our key observation is that, in analogy to the one-dimensional spin-Peierls transition, these terms may precipitate an instability. The response of $QED_3$ to this perturbation will be dominated by lattice distortions with crystal momentum $\mathbf{X}_a$.

While the irrelevant deformation (20) is a universal expression written down on symmetry grounds, it can be related to appropriate microscopic models. By using the mapping (17), we can see that the perturbation (20) is simply a nearest-neighbour Heisenberg coupling

$$H \sim J_1 \sum_i \sum_{a=1,2,3} \vec{S}_{\mathbf{R}_i} \cdot \vec{S}_{\mathbf{R}_i + \boldsymbol{\delta}_a}. \tag{25}$$

To implement the distorted lattice in this microscopic model, assume an exponential form the bond-dependent exchange $J(\mathbf{d}) = J_1 e^{-\alpha|\mathbf{d}|}$ which is independent of bond angle and where $J_1 = J(\boldsymbol{\delta}_a)$. A lattice distortion affects the Hamiltonian (25) by modifying the coupling constants $J(\mathbf{d}_{ij})$, which typically decrease as a function of distance $\mathbf{d}_{ij}$ between two magnetic ions. At $u = 0$ magnetic ions are at equilibrium (minimising the combination of magnetostriction and elastic energy cost on a given bond pair). We may then expand in small $u$, yielding a linear coupling between spin bilinears and the distortion field, with $\alpha \sim \partial J/\partial r|_{d_{ij}}$ as a constant of proportionality. More explicitly, taking $\mathbf{r}_i = \mathbf{R}_i + \mathbf{u}(\mathbf{r}_i)$ and expanding in small $|\mathbf{u}(\mathbf{r}_i)|/a$ gives the following spin-lattice coupling

$$H_\alpha[\mathbf{u}(\mathbf{r}_i)] \simeq (\alpha J_1/a) \sum_i \sum_{a=1,2,3} \boldsymbol{\delta}_a \cdot [\mathbf{u}(\mathbf{r}_i) - \mathbf{u}(\mathbf{r}_i + \boldsymbol{\delta}_a)] \vec{S}_{\mathbf{r}_i} \cdot \vec{S}_{\mathbf{r}_i + \boldsymbol{\delta}_a} \tag{26}$$

at first order. Here, $\alpha$ can be interpreted as a microscopic spin-lattice coupling parameter, which is non-trivially related to the field-theoretic monopole coupling $g$.

We model lattice distortions as having a potential energy that is quadratic in the relative displacement between nearest-neighbour sites,

$$H_{\text{ph}} = \frac{K}{2} \sum_{\langle ij \rangle} |\mathbf{u}(\mathbf{r}_i) - \mathbf{u}(\mathbf{r}_j)|^2. \tag{27}$$

Here, $K$ takes the role of an effective spring constant for the displacement between atoms on the triangular lattice. Taking the continuum limit, inserting (22), and decomposing into transverse and longitudinal polarisations $\mathbf{u_q} = \sum_{s=t,l} \boldsymbol{\epsilon}_{\mathbf{q},s} u_{\mathbf{q},s}$ yields the Hamiltonian density

$$\mathcal{H}_{\text{ph}}[\mathbf{u}] = \sum_{\mathbf{q}} \mathcal{K}_{\mathbf{q}} |\mathbf{u_q}|^2 = \sum_{\mathbf{q},s} \mathcal{K}_{\mathbf{q}} |u_{\mathbf{q},s}|^2,$$
$$\mathcal{K}_{\mathbf{q}} = \frac{2K}{3a^2} \left( 3 - \left[ \cos(aq_x) + 2\cos\left(\frac{aq_x}{2}\right) \cos\left(\frac{\sqrt{3}aq_y}{2}\right) \right] \right). \tag{28}$$

The polarisation vectors $\boldsymbol{\epsilon}_{\mathbf{q},s}$ are orthonormal, where the longitudinal direction is defined as $\boldsymbol{\epsilon}_{\mathbf{q},l} = \mathbf{q}/|\mathbf{q}|$. Importantly, the complex scalar modes $u_{\mathbf{q},s}$ are independent and degenerate, meaning the Hamiltonian of the longitudinal modes can be seperated.

## Conformal perturbation theory

Our goal is to study whether the phonon-monopole coupling generates instability. To this end, we compute the energy of $QED_3$ in the background of arbitrary displacement fields $\mathbf{u}(\mathbf{r})$ and extremise the resulting energy functional. It is convenient to employ a path-integral formulation, where we obtain a functional for the effective energy density $\mathcal{E} = E/V$ (per volume $V$) via the zero-temperature limit of the free energy,

$$\mathcal{E}_{QED_3}[\mathbf{u}] = \frac{1}{V} \lim_{\beta \to \infty} F_{QED_3}[\mathbf{u}]. \tag{29}$$

The free energy of $QED_3$ coupled to some background displacement field $\mathbf{u}(\mathbf{r})$,

$$F_{QED_3}[\mathbf{u}] = -\frac{1}{\beta} \log \mathcal{Z}_{QED_3}^{S_\beta^1 \times \Sigma}[\mathbf{u}], \tag{30}$$

is given in terms of the partition function

$$\mathcal{Z}_{QED_3}^{S_\beta^1 \times \Sigma}[\mathbf{u}] = \int \mathcal{D}[\{\mathcal{O}_{\text{CFT}}\}] \, e^{-S_{QED_3} - S_g[\mathbf{u}]} \tag{31}$$

with the respective actions $S_A = \int_0^\beta d\tau L_A$ on the manifold $S_\beta^1 \times \Sigma$. Here, $S_\beta^1$ corresponds to a circle in the imaginary time direction with circumference inverse temperature $\beta = 1/T$, and $\Sigma$ is some spatial manifold (for example a 2-sphere with radius $L$, $S_L^2$). We mostly focus on the thermodynamic limit $\Sigma = \lim_{L \to \infty} S_L^2 = \mathbb{R}^2$ (note that the partition function $\mathcal{Z} = \mathcal{Z}^{S_L^3}$ may also be defined on a 3-sphere with radius $L$, see, e.g., ref. 29, making the conformal SO(3, 2) symmetry group manifest).

In the limit of weak coupling we can work perturbatively and exploit the fact that two-point functions at the CFT fixed point are known. We assume that the monopole-lattice action is a small perturbation, justified by assuming small coupling $g$ and by our previous assumption that $|\mathbf{X}_a \cdot \mathbf{u_Q}| \ll 1$. Expanding the Boltzmann weight to quadratic order and taking the logarithm, we have

$$\log \mathcal{Z}_{QED_3}[\mathbf{u}] = \log \mathcal{Z}_{QED_3} - \langle S_g \rangle_{QED_3} + \frac{1}{2} \left( \langle S_g^2 \rangle_{QED_3} - \langle S_g \rangle_{QED_3}^2 \right) + \dots, \tag{32}$$

where the expectation values are to be taken with respect to the path integral (31) with $\mathbf{u} = 0$. At finite temperatures $\beta < \infty$, one-point functions of conformal primaries are generically non-zero[62] $\langle \mathcal{O} \rangle \sim \beta^{-\Delta_\mathcal{O}}$, but, importantly, vanish in the zero-temperature limit $\beta \to \infty$, as also mandated by conformal invariance on $\mathbb{R}^3$. Because we will primarily focus on the zero-temperature limit, we henceforth take $\langle S_g \rangle_{QED_3} = 0$.

The first non-trivial contribution to $\mathcal{E}_{QED_3}[\mathbf{u}]$ thus occurs at quadratic order (we use the notation $x = (\tau_x, \mathbf{r}_x)$ for vectors in 2+1-dim. Euclidean spacetime),

$$\langle S_g^2 \rangle_{QED_3} = g^2 \sum_{a,b} s_a s_b \sum_{\mathbf{Q},\mathbf{Q}'} \int_{S_\beta^1 \times \Sigma} d^3x \, d^3y$$
$$\left[ (i\mathbf{X}_a \cdot \mathbf{u_Q}) e^{i(\mathbf{X}_a + \mathbf{Q}) \cdot \mathbf{r}_x} (-i\mathbf{X}_b \cdot \mathbf{u}^*_{\mathbf{Q}'}) e^{-i(\mathbf{X}_b + \mathbf{Q}') \cdot \mathbf{r}_y} \langle \Phi_a(x) \Phi_b^\dagger(y) \rangle_{QED_3} + \text{h.c.} \right]$$
$$= g^2 \beta V \sum_a \int_0^\beta d\tau_{x'} \int_\Sigma d^2\mathbf{r}_{x'} \left[ |\mathbf{X}_a \cdot \mathbf{u_Q}|^2 e^{i(\mathbf{X}_a + \mathbf{Q}) \cdot \mathbf{r}_{x'}} \langle \Phi_a^\dagger(x') \Phi_a(0) \rangle_{QED_3} + \text{h.c.} \right], \tag{33}$$

where $V = \text{vol}(\Sigma) = 2\pi L^2$ is a trivial factor of volume, mandated by the extensiveness. The thermodynamic limit $\Sigma = \mathbb{R}^2$ is obtained by taking the linear dimension $L \to \infty$.

For $\mathbf{Q} \neq -\mathbf{X}_a$ there exist rapidly oscillating terms in Eq. (33) (with momentum $\mathbf{X}_a + \mathbf{Q}$), which, assuming that the monopole two-point function varies sufficiently slowly, produce finite contributions to $\langle S_g^2 \rangle_{QED_3}$ which average out at sufficiently long length scales (in the continuum limit). The term with $\mathbf{Q} = -\mathbf{X}_a$ in Eq. (33) hence is the dominant contribution at low energies, which is due to the (relevant) coupling between monopoles and displacements at lattice momentum $\mathbf{X}_a$, as was written before in Eq. (24).

As noted in ref. 62, the cylindrical geometry is conformally flat in the $L \to \infty$ limit, and OPEs converge for $\sqrt{\tau^2 + |\mathbf{r}^2|} < \beta \to \infty$, such that we

may use above form of the 2-pt. function to do the remaining integral. This integral is divergent in the thermodynamic, zero temperature limit, $\beta, L \to \infty$. At any other wavevector $\mathbf{Q} \neq -\mathbf{X}_a$, the exponential in (33) cuts off the divergent integral and no instability occurs. We regulate this divergence by working at finite temperatures $\beta < \infty$, which allows us to take the thermodynamic limit $L \to \infty$,

$$\frac{\langle S_g^2 \rangle_{\mathrm{QED}_3}}{\beta V} = 2g^2 \sum_a \int_0^\beta d\tau \int_\Sigma d^2\mathbf{r} \, \frac{|\mathbf{X}_a \cdot \mathbf{u}_{\mathbf{X}_a}|^2}{(\tau^2 + |\mathbf{r}|^2)^{\Delta_\Phi}} = c_{\Delta_\Phi} g^2 \beta^{3-2\Delta_\Phi} \sum_a |\mathbf{X}_a \cdot \mathbf{u}_{\mathbf{X}_a}|^2 \tag{34}$$

where we have introduced the numerical constant $c_{\Delta_\Phi} = 2\pi/[(\Delta_\Phi - 1)(3 - 2\Delta_\Phi)] > 0$ for $1 < \Delta_\Phi < 3/2$.

We can now obtain the effective energy as a sum of the intrinsic potential energy cost of deformations at wavevectors $\mathbf{X}_a$ and the relative energy gain of the DSL by coupling to the displacement field,

$$\begin{aligned}\mathcal{H}_{\mathrm{eff}}[\mathbf{u}] &= \mathcal{H}_{\mathrm{ph}}[\mathbf{u}] + \left( \mathcal{E}_{\mathrm{QED}_3}[\mathbf{u}] - \mathcal{E}_{\mathrm{QED}_3}[0] \right) \\ &= \sum_{a=1,2,3} \left( \mathcal{K}_{\mathbf{X}_a} |\mathbf{u}_{\mathbf{X}_a}|^2 - \lim_{\beta\to\infty} c_{\Delta_\Phi} g^2 \beta^{3-2\Delta_\Phi} |\mathbf{X}_a \cdot \mathbf{u}_{\mathbf{X}_a}|^2 \right) \end{aligned} \tag{35}$$

We observe that the second term in the parenthesis is minimised by taking lattice displacements that are longitudinal along the momenta $\mathbf{u}_{\mathbf{X}_a} = i s_a u_a \mathbf{X}_a/|\mathbf{X}_a|^2$. The scalar amplitudes $u_a$ are the longitudinal components $u_{\mathbf{X}_a,l}$ at the three independent displacement vectors $\mathbf{X}_a$ (up to a phase), and we introduce $\kappa = \mathcal{K}_{\mathbf{X}_a}/|\mathbf{X}_a|^2 \sim K$ to express Eqs. (24) and (35) compactly as Eqs. (2) and (3).

Now we turn to an alternative finite-system-size regularisation at $T = 0$. For a cylinder geometry with any finite circumference (smaller than the correlation length), the system's energy will be an analytic function in the spin-Peierls coupling $g$, and the change in energy $\Delta\mathcal{E}_{\mathrm{QED}_3}[\vec{u}] = \mathcal{E}_{\mathrm{QED}_3}[\vec{u}] - \mathcal{E}_{\mathrm{QED}_3}[0]$ due to coupling to a distortion field can be obtained using the same perturbative expansion,

$$\Delta\mathcal{E}_{\mathrm{QED}_3}[\vec{u}] = -\lim_{\beta\to\infty} \frac{1}{2\beta V} \langle S_g^2 \rangle_{\mathrm{QED}_3} + \ldots, \tag{36}$$

where $\langle S_g^2 \rangle_{\mathrm{QED}_3}$ can be obtained in an analogous manner to (34), but now performing the space-time integral on the geometry $\lim_{\beta\to\infty} S_\beta^1 \times S_L^2$, such that

$$\Delta\mathcal{E}_{\mathrm{QED}_3}[\vec{u}] = \tilde{c}_{\Delta_\Phi} g^2 L^{3-2\Delta_\Phi} |\vec{u}|^2 \tag{37}$$

where the constant $\tilde{c}_{\Delta_\Phi} = 2\pi^{3/2}\Gamma(\Delta_\Phi - 1/2)/[(3 - 2\Delta_\Phi)\Gamma(\Delta_\Phi)]$. For the small systems that can be simulated numerically, we expect to be in the weak-coupling regime, defined in the new regularisation scheme as $g^2 L^{3-2\Delta_\Phi} |\vec{u}| \ll 1$.

## Numerical study

We use the infinite density matrix renormalization group algorithm (DMRG)[63–66] to study the lattice model on a cylinder of finite circumference $L_y \equiv L$ [67,68] and infinite length $L_x$. This limits the scope of the simulation by the introduction of a finite-size gap to the gapless DSL, but we will find that strong signatures of the DSL response remain, as seen in a previous study of the dynamical structure factor on a cylindrical geometry[34,69]. We aim to provide supporting numerical evidence for our analytical results with a study of the $J_1$–$J_2$ triangular lattice in the DSL phase by demonstrating a response consistent with a weak coupling instability under lattice distortions. The exchanges are modified via the (dimensionless) distortion parameter $\delta = \alpha u$ for a given pattern, where $\alpha$ is a constant microscopic spin-lattice coupling. In all cases, the states can be written as either one or a sum of three momentum eigenstates satisfying the normalisation $\sum_{\mathbf{q}} |\mathbf{u}_{\mathbf{q}}|^2 \equiv u^2 = \delta^2/\alpha^2$; their explicit real-space forms are given in the Supplementary Materials. The

energy response of the system in the weak-coupling regime may be written $\mathcal{H}[\delta] = (\mathcal{K}_{\mathbf{Q}}/\alpha^2 - A_{\mathbf{Q}}^L)\delta^2$, where $A_{\mathbf{Q}}^L$ is the coefficient of the energy gain of the spins on a finite cylinder, to be computed numerically. The potential energy cost of the patterns are not necessarily equal, but through Eq. (28) we evaluate

$$\mathcal{K}_{\mathbf{M}} = \frac{8K}{3a^2}, \quad \mathcal{K}_{\mathbf{K}} = \frac{5K}{3a^2}, \quad \mathcal{K}_{\mathbf{X}} = \frac{(5 - 2\sqrt{3})K}{3a^2}. \tag{38}$$

The patterns with larger unit cells have a generally smaller energy cost for the same momentum-space distortion magnitude $u^2$, meaning the $\mathbf{X}_a$ patterns have the lowest potential energy cost of the patterns we will compare.

To model the lattice distortion, we stabilise the spin-disordered ground state on a translationally invariant lattice (we take the $J_2 = J_1/8$ parameter value in the spin-liquid regime). We simulate a cylindrical geometry infinite in the $x$-direction by repeating an $L_x = 3$, 6 unit cell (chosen to be compatible with the distortion pattern). We consider first a cylinder with circumference $L = 6$ using the YC6 boundary conditions[19,70]. We then introduce a small distortion with magnitude $\delta$ of the lattice according to one of the patterns, modify the NN and NNN bonds accordingly, and then use DMRG to find the ground state on the new lattice. We use the spin-disordered ground state as the initial state and make no assumptions about the resultant spin state. We proceed by increasing the distortion parameter $\delta$ by a small amount and calculating the resultant ground state with its energy. This process is repeated for increasing bond dimension $\chi$ (up to 4000) in order to ensure the resultant energy differences are well converged. The VBS correlations shown have been obtained for bond dimension $\chi = 2000$ after adiabatically increasing the distortion $\delta$. The undistorted ground state had been optimised using the odd-sector method for the same bond dimension (cf. Supplementary Note 4).

We consider finite-size systems with circumference $L_y = 3$, 6, 9; since the symmetric pattern cannot fit on these new geometries, we are restricted to only studying the momentum-eigenstate patterns $\mathbf{Q} = \mathbf{M}_3, \mathbf{K}_3, \mathbf{X}_3$. These patterns all fit in a six-site unit cell, which we can fit on the YCN geometry for $N$ a multiple of 3. The responses $\Delta E(\delta)$ for $L = 3$, 6 are well converged for all $\delta$ and well described by a quadratic (fitted to exponent 2); for $L = 9$, we are able to converge one point $\delta = 0.002$ for the three patterns (using up to $\chi = 7000$). Due to this numerical limitation, we cannot compare the amplitudes obtained from fitting for all system sizes, so instead we focus on comparing the energy gain $\Delta E(\delta = 0.002)$ in Fig. 2d as a function of system size.

## Dynamical phonons

In this section, we will evaluate the effect of algebraic VBS fluctuations of the stable DSL phase on the spectrum of the phonons. At zero-temperature we find a continuous spectrum of phonons[39] with a divergent spectral weight as $\omega \to 0$. At larger temperature, the analysis in the main text shows there should instead be a well-defined pole in the phonon propagator with an energy that corresponds to a renormalised phonon frequency.

To describe this effect, we first effectively incorporate the VBS-monopole fluctuations of the DSL to produce the dressed propagator of the (momentum-dependent) phonon modes at quadratic order[39],

$$G(\omega, \mathbf{k}) = \frac{1}{\omega_0(\mathbf{k})^2 - g^2\rho^{-1}\chi_{\mathrm{VBS}}(\omega, \mathbf{k}) - \omega^2} \tag{39}$$

where we take the bare phonon dispersion $\omega_0(\mathbf{k})^2 = \mathcal{K}_{\mathbf{k}}/(\rho|\mathbf{k}|^2)$, which is valid close to $\mathbf{X}_a$ where it remains gapped. The spin-VBS channel susceptibility is given by Eq. (19) at $T = 0$, featuring a divergent continuum of excitations at $\omega > c|\mathbf{k} - \mathbf{X}_a|$, but is not generally known at finite temperature. One can write generally $\chi_{\mathrm{VBS}}(\omega, \mathbf{k}) = T^{2\Delta_\Phi - 3} F(\omega/T,$

$c|\mathbf{k} - \mathbf{k}_a|/T)$, where $F$ is an unknown universal scaling function and constrain it in the large-temperature limit[42,71,72].

First, we derive the phase diagram by evaluating the correction to the $\mathbf{X}_a$ phonon energy $\omega_a^2$ via Eq. (15). For temperatures $T \gg \omega$, scaling arguments imply that to leading order $\chi_a'(\omega, T) \sim T^{-(3-2\Delta_\Phi)}(c + \mathcal{O}(\omega/T))$ with some constant $c$ [42], and from (15) it follows that the phonon frequency (pole of $G_a(\omega)$) is shifted downwards.

We can find an explicit solution to the equation (15) by considering $\omega \to 0$ and $T > 0$. In this case, above scaling form for $\chi_a'$ becomes exact, yielding $0 = \omega_0^2 - g^2\rho^{-1}\chi_a'(0, T)$. This is solved when $1 \sim g^2\kappa^{-1}T^{-(3-2\Delta_\Phi)}$, which precisely recovers the critical scaling of the spin-Peierls temperature $T_{\mathrm{SP}}$ as a function of $g$ in Eq. (4).

Conversely, we can analyse the zero-temperature $T \to 0$ limit in (15) working perturbatively: If the second term on the right-hand side of (15) is much smaller than first term, one can iteratively substitute (the square root of) the left-hand side for $\omega$ on the right-hand side, generating an order-by-order expansion. To leading order, we thus obtain

$$\omega^2 \approx \omega_0^2 - g^2\rho^{-1}\chi_a'(\omega_0, 0) + \dots. \tag{40}$$

Using $\chi_a'(\omega, 0) \sim \omega^{-(3-2\Delta_\Phi)}$, one obtains that the phonon dispersion is renormalised down to zero energy $\omega = 0$ when $1 \sim g^2\kappa^{-1}\omega_0^{-(3-2\Delta_\Phi)}$, which coincides with the parameter regime where the perturbative treatment of (15) breaks down, and yields the scaling law (13) for the critical spin-Peierls coupling at finite frequency and zero-temperature.

Now we can calculate the momentum-resolved phonon spectral function: it is given in terms of (39) by $S_{\mathrm{phonon}}(\omega, \mathbf{k}) = 2\operatorname{Im} G(\omega + i\epsilon, \mathbf{k})$. This second-order perturbative result is plotted using Eq. (19) at $T = 0$ in Fig. 3a. While even asymptotic results for the frequency-dependent susceptibility at finite-$T$ are not exactly known, for illustrative purposes we heuristically assume a form similar to the expansion obtained by Sachdev and Ye[71];

$$F^{-1}(x, y) = C + x^2 - \gamma y^2 + \cdots \tag{41}$$

where we choose $C = 0.3$ and $\gamma = 1$. In Fig. 3c, we plot the phonon spectral function in the large-$T$ regime which shows a temperature-dependent softening of the phonon mode towards zero at the spin-Peierls temperature. This is accompanied by the emergence of a sharp divergent continuum of excitations which blurs with the mode at low energies. A better understanding of the scaling form $F$ would allow a quantitative prediction of RIXS and neutron scattering experiments, as well as the full phase diagram.

## Data availability
The numerical data presented in this work are available on Zenodo[73].

## Code availability
The DMRG code is related to the publicly available TeNPy library[74] (both deriving from an earlier common version). The lattice-class implementation with distortions (also used for constructing the DMRG model), alongside analytic predictions for the phonon spectral function and the VBS order pattern, are available on Zenodo[73].

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

## Acknowledgements

We thank L. Balents, F. Becca, Z.-X. Luo, J. Schmalian, and A. Wietek for insightful discussions. We would like to especially thank F. Ferrari and R. Valentí for their ongoing collaboration, as well as discussions and results which helped to predict the symmetries of the VBS ordered phase.
U.F.P.S. acknowledges support from the Deutsche

Forschungsgemeinschaft (DFG, German Research Foundation) through a Walter Benjamin fellowship, Project ID 449890867. F.P. acknowledges support from the DFG through Germany's Excellence Strategy EXC-2111-390814868. J.K. acknowledges support from the DFG under Germany's Excellence Strategy EXC-2111-390814868, DFG grants No. KN1254/1-2, KN1254/2-1, and TRR 360 - 492547816. This research was supported by the National Science Foundation under Grant No. NSF PHY-1748958 and the European Research Council (ERC) under the European Union's Horizon 2020 research and innovation program (Grant Agreement No. 771537). J.K. also acknowledges the support of the Munich Quantum Valley, which is supported by the Bavarian state government with funds from the Hightech Agenda Bayern Plus.

## Author contributions

U.F.P.S. and J.W. contributed equally to this work. M.D. performed the numerical simulations. F.P. and J.K. supervised the project. All authors contributed equally to the analysis of the results and preparation of the manuscript.

## Funding

## Competing interests

The authors declare no competing interests.
