## [Peer Review File · Nature Communications]

Spin-Peierls instability of the U(1) Dirac spin liquidREVIEWER COMMENTS

Reviewer #1 (Remarks to the Author):

The manuscript “Spin-Peierls instability of the U(1) Dirac spin liquid” by Seifert et al. addresses a key outstanding issue on the monopole instabilities of Dirac spin liquids (DSL) in the presence of lattice distortions. This is a timely and thorough treatment of the problem which is of utmost theoretical and more so current experimental relevance in light of claims of realization of DSL phase in the triangular lattice compound $\text{YbZn}_2\text{GaO}_5$ [arXiv: 2305.20040 (2023)]. Broadly speaking, the methodological treatment phonon-monopole coupling is of a wider relevance given that ground states of spin-1/2 Heisenberg antiferromagnets on other 2D lattices also feature DSL as ground states, e.g., the kagome lattice.

The authors undertake a thorough treatment of the monopole-lattice coupling and the resulting spin-Peierls instability of the DSL employing complementary approaches based on perturbative and scaling analysis to address the weak- and strong-coupling regimes, respectively. This leads to their first important result pointing to an instability of the DSL in the static phonon approximation together with an estimate of the critical spin-Peierls temperature for the onset of the symmetry broken state and the magnitude of the induced displacement field. Subsequently, the authors predict the pattern of spin-spin correlations which leads to a translational symmetry broken

valence bond crystal ground state with a 12-site unit cell and C_3 point group symmetry. This result is of immense value as it holds a concrete predictive power for both numerical simulations and signatures in experiment. Indeed, a careful density matrix renormalization group analysis supplemented with finite-size scaling then follows, investigating all potential VBC patterns and indeed identifying the 12-site VBC as the one featuring the largest energy gain and size dependence, in striking consonance with the field theoretical predictions.

The most interesting part of the manuscript with the most exciting results of experimental relevance then follow via a treatment of dynamical phonons. Here, the authors argue and demonstrate that a finite phonon frequency prevents a weak coupling instability of the DSL and they determine the critical coupling (monopole-monopole) scale determining the transition up to which the DSL remains stable, which I believe is a central result of this paper. It certainly provides bounds which will prove crucial to assess the potential real-material realization of DSLs.

The analytical and numerical treatments are technically cutting-edge and of a thorough character leading to a water-tight case as regards the conclusions made. Moreover, the results are of prime significance in that they substantially further our understanding of spin liquids and given the intertwined character of the problem to other quantum fluids and the relevance of spin liquids in other related condensed

matter phenomena, I expect them to be of general interest to the condensed matter community. In essence, the authors have addressed a broad question pertaining to 2D analogue of the well-known spin-Peierls instabilities in 1D via treating an extremely pressing issue (stability of DSLs) which has been garnering progressively increasing attention from theoretical, numerical and experimental communities. The theoretical and numerical aspects are not just restricted to condensed matter physics but also straddle the boundaries with topics in high-energy physics and lattice QED which will definitely garner the attention of the wider physics community.

Thus, I strongly recommend the publication of this manuscript in Nature Communications.

Reviewer #2 (Remarks to the Author):

The authors have performed a study of spin-Peierls instability of the Dirac spin liquid (DSL), which has been theoretically and numerically predicted as a ground state for certain frustrated antiferromagnets in two dimensions. The first finding is a symmetry-allowed coupling between phonons at a particular lattice wavevector to a magnetic instanton of the infrared field theory theory (QED3), which corresponds by the results of Song et al. to an order parameter of a valence-bond solid (VBS) ordering tendency of the underlying spin model, leading to a weak-coupling instability where the VBS order is (presumably) induced along with a static lattice distortion.

The authors also provide supporting numerical evidence of this instability using cylinder DMRG for the J_1 - J_2 triangular lattice Heisenberg anti-ferromagnet, with results generally supportive of the prediction.

Lastly, the authors consider dynamical coupling of phonons to the DSL, and predict a critical coupling at which the instability appears, further demonstrating that the phonon

spectral function should exhibit a Kohn anomaly indicative of the deconfinement of the DSL spinous and the quantum-critical DSL phase, thus providing a novel experimental prediction in the search for compounds with a DSL ground state.

The findings in this work are both timely and significant. The DSL is a focus of ongoing theoretical, numerical, and experimental research. In any compound the role of dynamical phonons must be considered, and the prediction of such an instability, as well as an experimental signature of an incipient quantum spin liquid are both important advances in this field. The paper is well-written in a style which is readable and understandable, and we have no reason to doubt the validity of any of the above-summarised results reported by the authors. We therefore recommend the paper for publication in Nature Communications after the following minor points are addressed.

Suggested Improvements:

While the paper is generally very well-written, with an appropriate amount of context and with results clearly stated and conveyed via the figures, we have a few suggestions for improvement:

1. We are concerned that the "Strong-Coupling Behavior" section, the shortest section of the paper, is also the least-contextualized section. It appears that the purpose of this section is to derive general results which are independent of the particular system being considered. The authors seemingly report two predictions here: a universal ratio of $\Delta_{\text{gap}}/T_{\text{SC}}$, and a generalisation of this to generic gapless deconfined phases coupled to lattice distortions in any dimensions, followed by a statement that this unifies the spin-Peierls instability in 1D to the results of this paper. Clearly the authors are attempting to state some important observations here which place their results in a broader context, and we feel that this warrants a brief expansion, space permitting. The authors should state more clearly and precisely the points that they currently mostly allude to in the course of three sentences.

2. The authors do not mention or cite the follow-up results of Song et al. in Phys. Rev. X 10, 011033 (2020), which we find surprising. These results demonstrate that the monopole operators can be derived from the spinon band structure, providing an alternative route to the CFT methods used in the original paper and which have been heavily relied on in this paper. It would be worthwhile for the authors to at least comment on possible connections between the spin-Peierls instability and the spinon band topology results of Song et al., which seem likely to provide a more transparent perspective on the microscopic origin of the instability.

3. The authors omit any mention of acoustic phonons and consider optical phonons exclusively. While acoustic phonons are clearly uninteresting from a Peierls instability perspective, it should be clarified why they can be ignored when considering dynamical phonons.

Typos:

- line 102: “for for”

- line 133: I think it should read “finite (cylindrical) geometry”

- line 179: “the corresponding the relevant”

- SM line 191 “there may exist” x2

- Fig 3 caption: “(a) zero and (b) finite temperature” should be (b) and (c)

Reviewer #2 (Remarks on code availability):

The access to the code is restricted, “Access to the data will be granted upon reasonable request”. I don't consider this to be a desirable practice. Were the code readily available, I would have performed a cursory review, but with this barrier I omitted this step in the review, so as not to delay the process even more.

Reviewer #3 (Remarks to the Author):

Reviewer #4 (Remarks to the Author):

This manuscript presents an important generalisation of the renowned spin-Peierls to 2D, specifically focusing on algebraic spin liquids. Using field theoretic arguments and extensive numerics, the authors show how a weak-coupling instability between the monopoles of the emergent gauge field and lattice distortions leads to the appearance of valence-bond ordering in U(1) Dirac quantum spin liquids on the triangular and kagome lattice. The study thoroughly encompasses both static deformations as well as quantum mechanical phonons, and further discusses notable features in the phonon spectrum due to this coupling even in regimes where the quantum spin liquid may be stable, suggesting potential routes towards experimental detection of this elusive phase (or its proximity in the case of valence bond ordering).

Overall, the paper is technical but well structured and adequately clearly written. The results are of the seminal stature that I would expect to see published in Nature Communications, and I am happy to recommend that the paper be accepted. It will be of great interest to the theoretical and experimental communities working on magnetism, (emergent) gauge theories, and topological phenomena in general.

I have two major concerns that I would urge the authors to address.

1) in line 128, the authors assume a linear dependence of the exchange interactions on the distance between lattice sites. Usually exchange interactions depend exponentially on distance. What are the reasons underpinning this choice? How fundamental is it to the conclusions raised by the authors? Unless I am missing something obvious here, it would perhaps be appropriate to add some explanation around this statement in the manuscript.

2) in the caption of Fig.2, the authors point out that there is "show strong agreement up to the

exact distribution of weight inside the unit cell", which seems to me to be an important difference. Do the authors have an understanding of this? Would it be worth adding a paragraph in the main text to discuss this point further?

I then have several minor comments, mostly related to some carelessness with typos and notation. In a paper that is highly technical, careful proofreading and optimisation of the symbols can go a long way to make it more accessible to the unfamiliar reader.

typo in abstract: "temperature-dependent corrections to the phonon spectrum, which provideS"  provide

in line 82 there is "x=" after the right arrow in the inlined expression. I don't think it belongs there; if it does, then I do not understand the expression and perhaps further explanation may be beneficial

in line 84 (and other parts of the paper) the authors change from boldface "x" to non-boldface "x". Is this meant to be the difference between 2+1 vector notation and spatial vector notation?

line 85: with x and boldface x representing the spatiotemporal / spatial coordinates, I found the use of X for reciprocal space slightly confusing at first. Is there a reason for it?

line 87: \vec{u} is a vector and s_a is not. Maybe the notation for the latter could be made a little different? $s_a = +1, +1, -1$ for $a=1,2,3$?

the term in square bracket in Eq.(5) should be u_0 and not u , I think?

it would be helpful to say something about the role of α when it is used (without being introduced) in Eq.(6)

in the caption of Fig.2, I find the sentence "We explicitly take momentum eigenvalues K_3 , and all $M_{1,2,3}$ points" somewhat unclear. Is there a way to state it perhaps more accessibly? Also, the reference thereafter to a "shaded region" is confusing as the shaded region is barely visible in the figure. Could it be highlighted, perhaps with an arrow pointing the reader to it in the figure?

typo in line 102 "for for"

typo in line 122 and 129 "the the"

in line 140 I see the expression defining $\delta = \alpha a |\mathbf{u}|$. I presume "a" here is the lattice spacing? The same character has been used in nearby text -- and in the very same Eq.(6)! -- as an index $a=1,2,3$. I think it would help the reader to resolve this unnecessary ambiguity (and to define the lattice spacing here)

in line 152, do the authors mean X_a or X_3 ?

typo in 179: "the corresponding the relevant"

in the caption of Fig.3, "extending interacting phonon propagator" would read better with "... the interacting...", I think

same in line 233: "distortion in above systems" add "the"?

237: typo "infinitesimal"

Eq.24): missing arrow on top of second S vector

332: $u(x)$ may be missing boldface x instead of x . At least to be consistent with the text above Eq.(2). It would be good to clarify and cross check the notation across the manuscript, methods and supp. mat.

in Eq.(37), the character "a" appears in a font that has not been used elsewhere. I presume it's a typo?

occasionally, the notation "Figure. 3(c)" or "Table. 1" appears. I wonder if the "." may be a typo. See line 472 or line 136 in supp. mat.

line 84 in supp.mat. "a a function"  as

Response to Referee #1

The manuscript “Spin-Peierls instability of the U(1) Dirac spin liquid” by Seifert et al. addresses a key outstanding issue on the monopole instabilities of Dirac spin liquids (DSL) in the presence of lattice distortions. This is a timely and thorough treatment of the problem which is of utmost theoretical and more so current experimental relevance in light of claims of realization of DSL phase in the triangular lattice compound YbZn₂GaO₅ [arXiv: 2305.20040 (2023)]. Broadly speaking, the methodological treatment phonon-monopole coupling is of a wider relevance given that ground states of spin-1/2 Heisenberg antiferromagnets on other 2D lattices also feature DSL as ground states, e.g., the kagome lattice.

The authors undertake a thorough treatment of the monopole-lattice coupling and the resulting spin-Peierls instability of the DSL employing complementary approaches based on perturbative and scaling analysis to address the weak- and strong-coupling regimes, respectively. This leads to their first important result pointing to an instability of the DSL in the static phonon approximation together with an estimate of the critical spin-Peierls temperature for the onset of the symmetry broken state and the magnitude of the induced displacement field. Subsequently, the authors predict the pattern of spin-spin correlations which leads to a translational symmetry broken valence bond crystal ground state with a 12-site unit cell and C₃ point group symmetry. This result is of immense value as it holds a concrete predictive power for both numerical simulations and signatures in experiment. Indeed, a careful density matrix renormalization group analysis supplemented with finite-size scaling then follows, investigating all potential VBC patterns and indeed identifying the 12-site VBC as the one featuring the largest energy gain and size dependence, in striking consonance with the field theoretical predictions.

The most interesting part of the manuscript with the most exciting results of experimental relevance then follow via a treatment of dynamical phonons. Here, the authors argue and demonstrate that a finite phonon frequency prevents a weak coupling instability of the DSL and they determine the critical coupling (monopole-monopole) scale determining the transition up to which the DSL remains stable, which I believe is a central result of this paper. It certainly provides bounds which will prove crucial to assess the potential real-material realization of DSLs.

The analytical and numerical treatments are technically cutting-edge and of a thorough character leading to a water-tight case as regards the conclusions made. Moreover, the results are of prime significance in that they substantially further our understanding of spin liquids and given the intertwined character of the problem to other quantum fluids and the relevance of spin liquids in other related condensed matter phenomena, I expect them to be of general interest to the condensed matter community. In essence, the authors have addressed a broad question pertaining to 2D analogue of the well-known spin-Peierls instabilities in 1D via treating an extremely pressing issue (stability of DSLs) which has been garnering progressively increasing attention from theoretical, numerical and experimental communities. The theoretical and numerical aspects are not just restricted to condensed matter physics but also straddle the boundaries with topics in high-energy physics and lattice QED which will definitely garner the attention of the wider physics community.

Thus, I strongly recommend the publication of this manuscript in Nature Communications.

We thank the Referee for this extremely positive assessment of our results and the thoughtful discussion of the general significance of our work. We are very pleased to receive such a strong recommendation for publication.

Response to Referees #2 and #3

Note from referee #3:

Report from referee #2:

The authors have performed a study of spin-Peierls instability of the Dirac spin liquid (DSL), which has been theoretically and numerically predicted as a ground state for certain frustrated antiferromagnets in two dimensions. The first finding is a symmetry-allowed coupling between phonons at a particular lattice wavevector to a magnetic instanton of the infrared field theory (QED3), which corresponds by the results of Song et al. to an order parameter of a valence-bond solid (VBS) ordering tendency of the underlying spin model, leading to a weak-coupling instability where the VBS order is (presumably) induced along with a static lattice distortion.

The authors also provide supporting numerical evidence of this instability using cylinder DMRG for the J1-J2 triangular lattice Heisenberg anti-ferromagnet, with results generally supportive of the prediction.

Lastly, the authors consider dynamical coupling of phonons to the DSL, and predict a critical coupling at which the instability appears, further demonstrating that the phonon spectral function should exhibit a Kohn anomaly indicative of the deconfinement of the DSL spinons and the quantum-critical DSL phase, thus providing a novel experimental prediction in the search for compounds with a DSL ground state.

The findings in this work are both timely and significant. The DSL is a focus of ongoing theoretical, numerical, and experimental research. In any compound the role of dynamical phonons must be considered, and the prediction of such an instability, as well as an experimental signature of an incipient quantum spin liquid are both important advances in this field. The paper is well-written in a style which is readable and understandable, and we have no reason to doubt the validity of any of the above-summarised results reported by the authors. We therefore recommend the paper for publication in Nature Communications after the following minor points are addressed.

We thank the two Referees for this positive assessment and their enthusiastic recommendation for publication.

Suggested Improvements:

While the paper is generally very well-written, with an appropriate amount of context and with results clearly stated and conveyed via the figures, we have a few suggestions for improvement:

1. We are concerned that the "Strong-Coupling Behavior" section, the shortest section of the paper, is also the least-contextualized section. It appears that the purpose of this section is to derive general results which are independent of the particular system being considered. The authors seemingly report two predictions here: a universal ratio of $\Delta_{\text{gap}}/T_{\text{SC}}$, and a generalisation of this to generic gapless deconfined phases coupled to lattice distortions in any dimensions, followed by a statement that this unifies the spin-Peierls instability in 1D to the results of this paper. Clearly the authors are attempting to state some important observations here which place their results in a broader context, and we feel that this warrants a brief expansion, space permitting. The authors should state more clearly and precisely the points that they currently mostly allude to in the course of three sentences.

Thank you for the helpful suggestion. Our intention is to focus here on the complementary description of the instability, focusing on how this approach is applicable in the thermodynamic limit, where deforming the CFT leads to non-trivial power-law responses determined by the scaling dimensions of appropriate operators at the conformal fixed point. We have rewritten this section and added context that we hope makes this clear – describing the alternative way of seeing the instability, and then discussing our predictions for future numerical works and for experimental measurements of the gap/critical temperature ratio.

-
2. The authors do not mention or cite the follow-up results of Song et al. in Phys. Rev. X 10, 011033 (2020), which we find surprising. These results demonstrate that the monopole operators can be derived from the spinon band structure, providing an alternative route to the CFT methods used in the original paper and which have been heavily relied on in this paper. It would be worthwhile for the authors to at least comment on possible connections between the spin-Peierls instability and the spinon band topology results of Song et al., which seem likely to provide a more transparent perspective on the microscopic origin of the instability.

We thank the Referee for pointing out our inconsistent referencing. Given that the two works by Song et al. are complementary, we have now made sure to cite them together throughout the manuscript when discussing the monopole transformation properties, not just in the Methods section. We emphasize that our results only rely on the monopole quantum numbers itself, which may be determined either numerically or analytically, as shown by Song et al. The monopoles' non-trivial quantum numbers under onsite and lattice translation symmetries may be understood in an analogous manner to the charge-flux attachment captured by Chern-Simons terms when integrating out fermions in a topologically non-trivial band coupled to a U(1) gauge field. While at present it is not clear to us if one can draw a more direct/microscopic connection between the Wannier centers (which determine the band topology) and the Peierls instability itself, we agree with the Referee that the role of spinon band topology can be emphasized more strongly in our manuscript. We have added the following:

"The monopole operators Φ_b carry non-trivial quantum numbers under lattice symmetries, which have been determined in Refs. [20,25] by complementary numerical and spinon band topology-based analyses. The latter makes evident how non-trivial monopole quantum numbers follow from the distribution of gauge charges in the system's unit cell."

-
3. **The authors omit any mention of acoustic phonons and consider optical phonons exclusively. While acoustic phonons are clearly uninteresting from a Peierls instability perspective, it should be clarified why they can be ignored when considering dynamical phonons.**

In reality, the phonon mode we are coupling to is an acoustic phonon band at finite wavevector, because the monopole operators carry some finite lattice momentum. It is to be expected that rather generally (in the absence of fine tuning) phonons at finite momenta are gapped. The spin-Peierls transition is thus controlled by the finite phonon frequency (at the relevant lattice momentum) relative to the coupling strength. Our field-theoretic approach is valid at low energies and we consider small continuum momenta p (or, equivalently, fields at long distances x) near the finite lattice momenta of the monopoles. If p is sufficiently small (compared to the inverse of the microscopic lattice spacing), we can thus consider the phonon to be non-dispersive, which significantly simplifies our analysis. This is in line with previous studies of the 1D spin-Peierls instability with dynamical phonons.

Since this is an important step in the analysis, we have added the following explanation and references:

“While we explicitly treat in this section a model of non-dispersive (optical) phonons, we note that this is also an appropriate model for generic phonon bands away from the Γ -point, as we are explicitly interested at small momenta (compared to the inverse lattice spacing) around the finite lattice momentum of the singlet monopoles \mathbf{X}_a .

[...]

The physical phonon modes have energy $\omega_0 \equiv \sqrt{\kappa/\rho}$ at the momentum \mathbf{X}_a ; we approximate this as an optical phonon with constant energy [5, 41], governed by the action...”

Typos:

- **line 102: “for for”**
- **line 133: I think it should read “finite (cylindrical) geometry”**
- **line 179: “the corresponding the relevant”**
- **SM line 191 “there may exist” x2**
- **Fig 3 caption: “(a) zero and (b) finite temperature” should be (b) and (c)**

Thank you, these have been fixed.

Remarks on code availability:

The access to the code is restricted, “Access to the data will be granted upon reasonable request”. I don’t consider this to be a desirable practice. Were the code readily available, I would have

performed a cursory review, but with this barrier I omitted this step in the review, so as not to delay the process even more.

We have made all data and plotting scripts freely available, along with the scripts for the analytical results of the manuscript (Figs. 2e, 3b & 3c). We have added corresponding code- and data-availability statements to the manuscript.

<https://zenodo.org/records/12725048>

Response to Referee #4

This manuscript presents an important generalisation of the renowned spin-Peierls to 2D, specifically focusing on algebraic spin liquids. Using field theoretic arguments and extensive numerics, the authors show how a weak-coupling instability between the monopoles of the emergent gauge field and lattice distortions leads to the appearance of valence-bond ordering in U(1) Dirac quantum spin liquids on the triangular and kagome lattice. The study thoroughly encompasses both static deformations as well as quantum mechanical phonons, and further discusses notable features in the phonon spectrum due to this coupling even in regimes where the quantum spin liquid may be stable, suggesting potential routes towards experimental detection of this elusive phase (or its proximity in the case of valence bond ordering).

Overall, the paper is technical but well structured and adequately clearly written. The results are of the seminal stature that I would expect to see published in Nature Communications, and I am happy to recommend that the paper be accepted. It will be of great interest to the theoretical and experimental communities working on magnetism, (emergent) gauge theories, and topological phenomena in general.

We thank the referee for their assessment that the results are of a seminal standing and their recommendation for publication.

I have two major concerns that I would urge the authors to address.

1. in line 128, the authors assume a linear dependence of the exchange interactions on the distance between lattice sites. Usually exchange interactions depend exponentially on distance. What are the reasons underpinning this choice? How fundamental is it to the conclusions raised by the authors? Unless I am missing something obvious here, it would perhaps be appropriate to add some explanation around this statement in the manuscript.

This is a case of imprecise wording, thank you for highlighting the confusion caused. We of course do take the natural exponential dependence on the bond length. When expanded for small distortions about the undistorted bond lengths, this gives the $(1-\alpha u+\dots)$ contribution that we study. We have changed the main text to be more clear, and additionally added a discussion of this approximation to the Methods.

“To this end, we assume that the couplings $J_{\{ij\}}$ between two sites are homogenous and decreasing functions with distance $|\vec{r}_i - \vec{r}_j|$, and we consider a simple exponential form $J_{\{ij\}} \sim J e^{-|\vec{r}_i - \vec{r}_j|/\alpha}$. In general, we expect that the first derivative of $J_{\{ij\}}$ as a function of distance is non-vanishing, which implies that for small distortions one may linearise.

[...]

A lattice distortion affects the Hamiltonian (25) by modifying the coupling constants $J(\vec{d}_{\{ij\}})$, which typically decrease as a function of distance $|\vec{d}_{\{ij\}}|$ between two magnetic ions. At $u=0$ magnetic ions are at equilibrium (minimizing the combination of

magnetostriction and elastic energy cost on a given bond pair). We may then expand in small u , yielding a linear coupling between spin bilinears and the distortion field, with $\alpha \sim \left. \frac{\partial J}{\partial r} \right|_{d_{ij}}$ as a constant of proportionality.”

2. **in the caption of Fig.2, the authors point out that there is "show strong agreement up to the exact distribution of weight inside the unit cell", which seems to me to be an important difference. Do the authors have an understanding of this? Would it be worth adding a paragraph in the main text to discuss this point further?**

Thank you for the helpful suggestion; we have taken this opportunity to explain more thoroughly in the main text the nature of the theoretical and numerical calculations. We would like to emphasize that the (universal) field-theoretic arguments predict the symmetry of the distortion but not the precise weight of $\langle S_i \cdot S_j \rangle$ on a given bond. This can be understood in an analogous manner to problems in superconductivity, where the linearised gap equation allows one to extract the transition temperature and the symmetry of the gap, but not the magnitude of the gap itself. The latter depends on microscopic details and (in principle) requires knowledge of the Landau Free Energy to arbitrarily higher order.

In order to clarify these points, we moved the explanation of how the (symmetry of) the distortion pattern was computed from the supplemental material into the main text and added a clarifying comment (see the updated Sec. ‘weak-coupling instability’).

“We emphasise that the computed patterns above predict the symmetry of the distorted lattice, but the precise strength of $\langle S_i \cdot S_j \rangle$ on a given bond is not accessible within our field-theoretic approach and is expected to depend on microscopic details.”

L235: “This leading-order mapping neglects possible multi-spin terms which may have the same transformation properties”

We additionally moved the sentence in question from the caption to the main text and rewrote it as

“The variation of singlet correlations in the distorted model shows good agreement relative to the prediction based on the monopole transformation properties in Fig. 2e where notably we see that the correlations are enhanced on all shortened nearest-neighbour bonds. The difference in the exact distribution of weight inside the unit cell is either attributable to neglecting contributions to the nearest-neighbour correlator by multi-spin operators, or errors introduced due to the use of an anisotropic $L = 6$ cylinder geometry”

I then have several minor comments, mostly related to some carelessness with typos and notation. In a paper that is highly technical, careful proofreading and optimisation of the symbols can go a long way to make it more accessible to the unfamiliar reader.

- **typo in abstract: "temperature-dependent corrections to the phonon spectrum, which provideS"  provide**

Thank you, this has been fixed.

- **in line 82 there is "x=" after the right arrow in the inlined expression. I don't think it belongs there; if it does, then I do not understand the expression and perhaps further explanation may be beneficial**

We confirm that the “ $x=$ ” is not misplaced, but instead is required to show that we are working with Eulerian coordinates where the distortion field is implicitly defined with respect to the global coordinate frame (in line with our response to the referee’s next question the expression now reads $\mathbf{r} = \mathbf{u}(\mathbf{r}) + \mathbf{R}$). The advantage of using Eulerian coordinates (rather than Lagrangian coordinates, wherein the displacement field is defined with respect to the undistorted lattice) consists in the fact that it preserves locality and allows for a natural and stringent continuum limit: writing local coupling of the form $O(\mathbf{r}) \mathbf{u}(\mathbf{r})$, where $O(\mathbf{r})$ is some field-theory operator, imply that $O(\mathbf{r})$ is attached to coordinate \mathbf{r} in the lab frame, which corresponds to a location in the distorted lattice. Conversely, within Lagrangian coordinates, a naive coupling of the form $O(\mathbf{R}) \mathbf{u}(\mathbf{R})$ is not sensible as \mathbf{R} refers to a coordinate of the old (undistorted) lattice, to which we no longer can attach a field theory operator. To clarify this, we have added the following comment when the displacement field is first introduced:

“Note that here we are using Eulerian coordinates [29] in which the displacement field is implicitly defined with respect to global (lab) coordinates (rather than with respect to the undistorted lattice) to preserve locality, ensuring a well-defined continuum limit.”

- **in line 84 (and other parts of the paper) the authors chance from boldface "x" to non-boldface "x". Is this meant to be the difference between 2+1 vector notation and spatial vector notation?**
- **line 85: with x and boldface x representing the spatiotemporal / spatial coordinates, I found the use of X for reciprocal space slightly confusing at first. Is there a reason for it?**

We have updated the conventions such that spatial positions are denoted by bold \mathbf{r} (and \mathbf{r}_i for lattice positions). We define in the text (L83) that $x = (\tau, \mathbf{r})$. The capital \mathbf{X}_a for reciprocal vectors is to follow a convention for naming this important $\mathbf{K}_a/2$ point [see fig 1 of PRX 14, 021010 (2024)].

- **line 87: \vec{u} is a vector and s_a is not. Maybe the notation for the latter could be made a little different? $s_a = +1, +1, -1$ for $a=1,2,3$?**
- **the term in square bracket in Eq.(5) should be u_0 and not u , I think?**

These inconsistencies have been fixed.

-
- **it would be helpful to say something about the role of alpha when it is used (without being introduced) in Eq.(6)**

We realize that we have been using the character α for two distinct quantities: previously, it referred to the microscopic spin-lattice coupling - as in Eq. (6) - and an exponent which we numerically fit to be 2.0 (in Line 140). We now give the latter exponent directly (without defining a variable), and have added the clarifying comment:

"Here, the (dimensionful) coefficient a is some constant of proportionality that arises upon linearising and characterizes the degree of spin-lattice coupling in the microscopic Heisenberg model."

-
- **in the caption of Fig.2, I find the sentence "We explicitly take momentum eigenvalues K_3 , and all $M_{1,2,3}$ points" somewhat unclear. Is there a way to state it perhaps more accessibly? Also, the reference thereafter to a "shaded region" is confusing as the shaded region is barely visible in the future. Could it be highlighted, perhaps with an arrow pointing the reader to it in the figure?**

The three M -points are symmetry inequivalent on the cylinder, and so we explicitly checked all three responses. This showed negligible dependence of the amplitude on orientation; while we chose to highlight this with a shaded region, we would like to refrain from drawing attention to a part of the figure with no physical content. We now appropriately refer to the Supplementary Materials where Fig1 shows the results for each orientation separately.

"Patterns considered are generated by momenta K_3 , and $M_{1,2,3}$ (here the purple shaded region represents the range responses for the three M_a , showing minimal dependence on cylinder orientation; see Supplementary Material D), as well as the 'full' 12-site distortion, defined by Eq. (5)."

-
- **typo in line 102 "for for"**
 - **typo in line 122 and 129 "the the"**
 - **in line 140 I see the expression defining $\delta = \alpha a |u|$. I presume "a" here is the lattice spacing? The same character has been used in nearby text -- and in the very same Eq.(6)! -- as an index $a=1,2,3$. I think it would help the reader to resolve this unnecessary ambiguity (and to define the lattice spacing here)**

Thank you, these typos have been fixed.

In this definition, the lattice spacing should not have been there; 'a' (in this font to mean lattice spacing) is only used in the Methods section and is appropriately defined.

- in line 152, do the authors mean X_a or X_3 ?

The calculation was performed on X_3 but we are attempting to make a statement about every X_a where the monopoles can couple. Given the limitations of the $L=9$ cylinder geometry, we may not assess this directly, but other calculations show the anisotropic geometry does not lead to strong splitting between the M_a points.

"We conclude that the numerical simulations performed on finite-circumference cylinders are compatible with our predictions that the instability (in large systems) will be dominated by distortions at the spin-singlet monopole wavevectors X_3 ."

- typo in 179: "the corresponding the relevant"
- in the caption of Fig.3, "extending interacting phonon propagator" would read better with "... the interacting...", I think
- same in line 233: "distortion in above systems" add "the"?
- 237: typo "infinitesimal"
- Eq.24): missing arrow on top of second S vector
- 332: $u(x)$ may be missing boldface x instead of x . At least to be consistent with the text above Eq.(2). It would be good to clarify and cross check the notation across the manuscript, methods and supp. mat.

Thank you, we have made these changes. We have also checked that all instances of 'x' as a parameter refer to space and time dependence.

- in Eq.(37), the character "a" appears in a font that has not been used elsewhere. I presume it's a typo?

This is intended to be the lattice spacing and is now used consistently through the Methods section. We appreciate the added complexity of overloading a single letter, but want to keep using 'a' as the index (the convention of Song et al., and the greek α as an index is commonly understood to only indicate x,y,z or $1,2,3$). We hope the different font makes it clear for expert readers who are interested in studying the Methods section.

- occasionally, the notation "Figure. 3(c)" or "Table. 1" appears. I wonder if the "." may be a typo. See line 472 or line 136 in supp. Mat.
- line 84 in supp.mat. "a a function"  as

These changes have also been made.

Thank you once again to the Referee for such a careful and detailed reading of the manuscript.